# Enhancing Optimizer Stability:
# Momentum Adaptation of The NGN Step-size

## Abstract

Modern optimization algorithms that incorporate momentum and adaptive step-size offer improved performance in numerous challenging deep learning tasks. However, their effectiveness is often highly sensitive to the choice of hyper-parameters, especially the learning rate. Tuning these parameters is often difficult, resource-intensive, and time-consuming. Therefore, recent efforts have been directed toward enhancing the stability of optimizers across a wide range of hyper-parameter choices (Schaipp et al., 2024). In this paper, we introduce an algorithm that matches the performance of state-of-the-art optimizers while improving stability through a novel adaptation of the NGN step-size method (Orvieto & Xiao, 2024). Specifically, we propose a momentum-based version (NGN-M) that attains the standard convergence rate of $\mathcal{O}(1/\sqrt{K})$ under common assumptions, without the need for interpolation condition or assumptions of bounded stochastic gradients or iterates, in contrast to previous approaches. Additionally, we empirically demonstrate that the combination of the NGN step-size with momentum results in high robustness while delivering performance that is comparable to or surpasses other state-of-the-art optimizers.

## 1 Introduction

Adaptive methods such as Adam (Kingma & Ba, 2015) and RMSprop (Hinton et al., 2012) are widely used in machine learning due to their established advantages over (momentum) SGD, particularly in tasks such as training Transformers (Brown, 2020; Touvron et al., 2021; 2023). These methods adaptively scale the step-size across different dimensions (parameters) based on their respective statistics, effectively acting as a diagonal precondition.

Although these methods perform well in practice, existing theoretical analyses typically require stringent assumptions on the noise structure of the stochastic gradients, such as sub-Gaussian noise (Li et al., 2024) or affine noise models (Wang et al., 2024; Zhang et al., 2024a). Relaxing these assumptions remains an open challenge. Another well-known issue with Adam is its sensitivity to the step-size hyper-parameter (Wilson et al., 2017; Choi et al., 2019), particularly when training Transformers, where loss spikes are commonly observed (Molybog et al., 2023; Wortsman et al., 2023). This often necessitates careful adjustments of the hyper-parameters throughout the training process (Zhang et al., 2022; Chowdhery et al., 2023), which can be costly in terms of computational resources (Or et al., 2020). Consequently, there has been growing interest in developing optimization methods that are more robust to hyper-parameter selection (Schaipp et al., 2024). In addition to adapting the step-size, Adam and other state-of-the-art optimizers also rely on momentum (Polyak, 1964), a broadly used technique that has been shown to enhance performance both theoretically (Cutkosky & Mehta, 2020; Fatkhullin et al., 2024; Islamov et al., 2024) and practically (Choi et al., 2019; Fu et al., 2023; Jelassi & Li, 2022). Besides speeding up convergence, momentum is known as a technique to reduce the variance of stochastic algorithms (Ma & Yarats, 2018; Cutkosky & Orabona, 2019), improving stability as well as generalization in some settings (Jelassi & Li, 2022).

In this work, we address the aforementioned drawbacks of Adam by developing a new algorithm based on the recently proposed NGN step-size (Orvieto & Xiao, 2024), an improved variant of the Stochastic Polyak Step-size (Loizou et al., 2021), that has demonstrated strong resilience to step-size hyper-parameter tuning: in particular, the algorithm was shown to never diverge for any choice

of the step-size hyper-parameter in the convex setting, and to exhibit strong curvature adaptation properties strengthened by theoretical guarantees. However, the step-size of Orvieto & Xiao (2024) simply adapts the learning rate through a scalar multiplier, leaving to future work the incorporation of momentum and coordinate-wise variants – needed in complex problems such as optimizing transformers, as motivated above. Here, we develop a momentum and step-size adaptive version of NGN designed to enhance robustness in terms of hyper-parameter selection. We also present a theoretical analysis alongside a practical evaluation of this approach, showcasing its improvements over current state-of-the-art methods.

In summary, our contributions are as follows:

1. First, we introduce a new algorithm named NGN-M that combines the NGN step-size with momentum. We theoretically show that NGN-M achieves a convergence rate $\mathcal{O}(1/\sqrt{K})$ in the convex regime without the typical requirements of interpolation or bounded gradient assumptions found in earlier works.

2. Next, we focus on the problem of adapting the step-size rule towards a coordinate-wise diagonal preconditioning. By integrating this diagonal step-size strategy with momentum, we develop a new variant of NGN, called NGN-MD.

3. The theoretical results are supported by extensive empirical validation in various deep learning settings where we demonstrate that NGN-M and NGN-MD not only preserve the robustness property of the NGN step-size, but improve it further in many cases. The step-size hyper-parameter resilience comes together with better performance comparable to that of state-of-the-art algorithms.

## 2 RELATED WORKS

**Polyak Step-size.** When training a deep network with standard optimizers, tuning the learning rate is crucial but time-consuming and resource-intensive (Goodfellow et al., 2016). This issue is at the root of recent research focusing on transferring hyper-parameters across architectures at different scales, therefore avoiding expensive tuning pipelines (Yang et al., 2022; 2023; Bordelon et al., 2023). Yet, already in the convex setting choosing the learning rate can be difficult – an issue that was studied already in Polyak (1987) and gave rise to the first adaptive method: the Polyak Stepsize (PS). Recently, there has been a renewed interest adapting PS to modern settings (Loizou et al., 2021; Orvieto et al., 2022; Jiang & Stich, 2024), delivering a theoretically principled way to adaptively scale the gradient magnitude during training. PS-inspired methods have gained increasing interest for their simplicity and adaptability, as they utilize local curvature and smoothness information to accelerate algorithms and facilitate faster convergence. Orvieto & Xiao (2024) recently introduced a variant of the Stochastic Polyak step-size, called NGN, which further enhances the robustness of the step-size hyper-parameter and solidifies the link to Gauss-Newton preconditioning. The theoretical analysis in Orvieto & Xiao (2024) demonstrated that NGN does not diverge regardless of the choice of the step-size hyper-parameter, and converges fast when the step-size is appropriately tuned. In contrast, the current theory of the SPS step-size with fixed step-size hyper-parameters (Loizou et al., 2021) proves convergence to the exact solution only if the interpolation condition holds[1].

**Polyak Step-size and Heavy-ball Momentum.** Heavy-ball momentum methods, stemming from the work of Polyak (1964), have gained significant attention over the years due to their benefits, including acceleration on convex quadratics (Jain et al., 2018; Lee et al., 2022; Bollapragada et al., 2022), convex-like (Wang et al., 2022), and non-convex problems (Cutkosky & Mehta, 2020), as well as their variance reduction abilities (Ma & Yarats, 2018; Cutkosky & Orabona, 2019). This has led to growing interest in the combination of Polyak step-size and heavy-ball momentum, which is now an active area of research (Barré et al., 2020; Saab et al., 2022; Barré et al., 2020; Wang et al., 2023; Oikonomou & Loizou, 2024). Recently, Schaipp et al. (2024) demonstrated that a geometrically principled combination of SPS and momentum leads to lower sensitivity to the step-size hyper-parameter, although they did not provide strong theoretical convergence guarantees.

**Diagonal Polyak Step-size.** Coordinate-wise adaptive step-sizes are essential in training Transformer architectures due to the varying parameter-wise scaling and conditioning of the problem

---

[1]In our notation, this means that $\sigma_{\text{int}}^2 = 0$.

Table 1: Summary of existing methods exploiting Polyak-type adaptive step-sizes and their convergence guarantees. **Mom.**=Supports momentum; **Diag.**=Supports diagonal step-sizes. $\sigma_{\text{int}}^2$ is defined in Section 4. $\mathcal{O}$ notation hides absolute and problem-dependent constant factors and logarithmic terms in the rate.

| Method | Rate [(a)] | Mom. | Diag. | Comments |
|---|---|---|---|---|
| SPS$_{\text{max}}$ (Loizou et al., 2021) | $\mathcal{O}(1/K + \sigma_{\text{int}}^2)$ | ✗ | ✗ | Conv. to non-vanishing neighbourhood |
| ALR-SMAG (Wang et al., 2023) | $\mathcal{O}((1-\rho)^K + \sigma_{\text{int}}^2)$ | ✗ | ✗ | Strong convexity Conv. to non-vanishing neighbourhood |
| Momo (Schaipp et al., 2024) | $\mathcal{O}(1/\sqrt{K})$ | ✗ | ✗ | Bounded stoch. gradients Interpolation |
| Momo-Adam (Schaipp et al., 2024) | ✗ | ✓ | ✓ | Momo framework for Adam |
| MomSPS$_{\text{max}}$ (Oikonomou & Loizou, 2024) | $\mathcal{O}(1/K + \sigma_{\text{int}}^2)$ | ✓ | ✗ | Conv. to non-vanishing neighbourhood |
| NGN (Orvieto & Xiao, 2024) | $\mathcal{O}(1/\sqrt{K})$ | ✗ | ✗ | – |
| NGN-M (Alg. 1) **[This work]** | $\mathcal{O}(1/\sqrt{K})$ | ✓ | ✗ | – |
| NGN-D (Alg. 3) **[This work]** | $\mathcal{O}(1/\sqrt{K})$ | ✗ | ✓ | – |
| NGN-MDv1 (Alg. 2) **[This work]** | ✗ | ✓ | ✓ | Combination of NGN-M and RMSprop |
| NGN-MDv2 (Alg. 2) **[This work]** | ✗ | ✓ | ✓ | Combination of NGN-M and NGN-D |

$(a)$ We report the convergence rates in one of three settings – strongly convex, convex, or non-convex – based on the results provided in the original paper where the respective method was introduced.

(Oikonomou & Loizou, 2024) provides two other combinations of SPS and momentum named MomDecSPS and MomAdaSPS. However, their convergence guarantees are derived in a setting with decreasing step-sizes and under a bounded iterates assumption, which makes them less favorable in practice.

(Noci et al., 2022; Zhang et al., 2024b). Algorithms employing diagonal step-sizes, such as Adam and Sign SGD (Bernstein et al., 2018), typically outperform non-diagonal methods in language modeling tasks by also addressing issues such as class imbalance (where certain words appear more frequently than others) (Kunstner et al., 2023; 2024) and heavy-tailed noise (Zhang et al., 2019; 2020). It is, therefore, paramount in current setups to deliver adaptive step-size improvements targeted to the coordinate-wise (diagonal) regime. However, most Polyak-step-size-based algorithms only focus on a single step-size for all parameters (Loizou et al., 2021; Wang et al., 2023; Gower et al., 2021; Oikonomou & Loizou, 2024; Orvieto & Xiao, 2024). Only a few works propose a diagonal-wise modification of Polyak-step-size by either using Adam preconditioner (Schaipp et al., 2024) as a weight matrix or incorporating second-order information from the objective function (Li et al., 2022; Richtárik et al., 2024).

Table 1 provides a theoretical comparison of various Polyak step-size-based algorithms that incorporate momentum and/or diagonal step-size, highlighting the differences between the theoretical results presented in this work and those from prior works.

## 3 ALGORITHM DESIGN OF NGN-M AND NGN-D

The NGN step-size, introduced by Orvieto & Xiao (2024), is derived by applying the Gauss-Newton method to the regularized Taylor expansion of the composition of a square and a square root of the positive-valued objective function defined as follows:

$$x^{k+1} = x^k + p^k \text{ where } p^k := \arg\min_{p \in \mathbb{R}^d} \left[ f_c(x^k + p) := (r(x^k) + \nabla r(x^k)^\top p)^2 + \frac{1}{2c}\|p\|^2 \right], \quad (1)$$

and $r(x) := \sqrt{f(x)}$. It turns out that the problem in (1) has a closed-form solution

$$p^k = -\gamma_k \nabla f(x^k) \text{ where } \gamma_k := \frac{c}{1 + \frac{c}{2f(x^k)}\|\nabla f(x^k)\|^2}$$

with $\gamma_k$ representing the NGN step-size. In Orvieto & Xiao (2024), convergence guarantees were established for both convex and general non-convex settings. Importantly, the convex analysis shows that NGN exhibits a non-divergence property, regardless of the step-size hyper-parameter $c$ (see Theorem 4.5 in Orvieto & Xiao (2024)). Due to this property, the NGN step-size is a strong candidate to achieve better robustness w.r.t. the choice of the step-size.

### 3.1 How to Add Momentum and What to Expect from It?

There are several approaches to combining the adaptive Polyak-type step-size with heavy-ball momentum. Broadly, existing algorithms can be divided into two categories: the first category involves computing the Polyak step-size in the usual manner and incorporating it into the standard heavy-ball update (Oikonomou & Loizou, 2024). In contrast, algorithms from the second category first determine an update direction using exponential weighted averaging of the stochastic gradient and momentum variable, and then compute the Polyak-type step-size based on the computed direction (Wang et al., 2023; Schaipp et al., 2024). Following this reasoning, we test two possible versions for combining the NGN step-size and momentum:

$$\text{Ver.1}: \begin{cases} \gamma_k = \frac{c}{1 + \frac{c}{2f(x^k)}\|\nabla f_{S_k}(x^k)\|^2} \\ m^k = \beta m^{k-1} + (1-\beta)\gamma_k \nabla f_{S_k}(x^k) \\ x^{k+1} = x^k - m^k \end{cases} \quad \text{Ver.2}: \begin{cases} m^k = \beta m^{k-1} + (1-\beta)\nabla f_{S_k}(x^k) \\ \gamma_k = \frac{c}{1 + \frac{c}{2f(x^k)}\|m^k\|^2} \\ x^{k+1} = x^k - \gamma_k m^k \end{cases}$$

Before we proceed, we should answer the question: "What do we expect from the combination of NGN step-size and momentum?" First, we aim to preserve, and ideally enhance, NGN's robustness to the step-size hyper-parameter. Additionally, we seek improved performance, achieving accelerated convergence akin to the advantage of SGD with momentum (SGDM) over standard SGD in convex settings. With these goals in mind, we now show that version 1 meets all of these criteria, while version 2 is less suitable. To gain some intuition regarding the performance of these two variants, we start by conducting a simple experiment on a quadratic function $f(x) = \frac{1}{2}\|Ax - b\|^2$ where $A$ is a data matrix from the normalized Diabetes dataset (Smith et al., 1988) and $b$ is a vector of labels. Based on the results from Figure 1 (left), we observe that variant 1 achieves accelerated convergence as SGDM for middle-range step-size hyper-parameters ($c \in \{10^1, 10^2\}$) and does not diverge for large step-size parameter ($c \in \{10^3\}$). Conversely, version 2 has a worse convergence rate than version 1 for middle-range step-size parameters and diverges for large ones. Therefore, we theoretically analyze and practically test version 1, which we call NGN-M.

### 3.2 Evidence of Robustness of NGN-M

One indication of the step-size resilience properties of NGN-M lies in the sharpness of the point where it converges. To illustrate this, we provide a simple example of minimizing a function $f(x) = (\sin(1+\cos(-\pi+x)) - 0.2x)^2 + (\sin(1+\cos(\pi-x)) + 0.2x)^4$ that has many sharp sub-optimal local and flat global minima. We compare the performance of NGN-M and SGDM varying the step-size hyper-parameter in $\{10^0, 10^1, 10^2, 10^3\}$ and the starting point in $[-20, 20]$ with a step $4/30$[^2]. Based on the results in Figure 1, we conclude that $(i)$ for small step-sizes, both methods likely get stuck at sub-optimal local minima and reach the global minima only if they are initialized close enough to it; $(ii)$ for large step-sizes, we observe less runs of SGDM reaching the global minima; $(iii)$ in contrast, for NGN-M with large step-sizes, we observe more runs reaching the global minima. This is possible due to the adaptive nature of the NGN step-size to the flatness of the global minima. Later, in Section 5 we demonstrate a similar convergence behavior of NGN-M when training a Resnet20 model.

### 3.3 Diagonal Step-size for NGN

To derive a diagonal version of NGN we modify an approach of (1). The next iterate $x^{k+1}$ is obtained by minimizing an approximation of the regularized first-order Taylor expansion of $r(x) \coloneqq \sqrt{f(x)}$ around $x^k$, namely, $x^{k+1} = x^k + p^k$ where

$$p^k = \underset{p \in \mathbb{R}^d}{\arg\min} \left[ f_{\boldsymbol{\Sigma}_k}(x^k + p) \coloneqq (r(x^k) + \nabla r(x^k)^\top p)^2 + \frac{1}{2c}\|p\|_{\boldsymbol{\Sigma}_k}^2 \right], \tag{2}$$

[^2]: This step is chosen small enough so that the initial point can be close to any local minima within $[-20, 20]$.

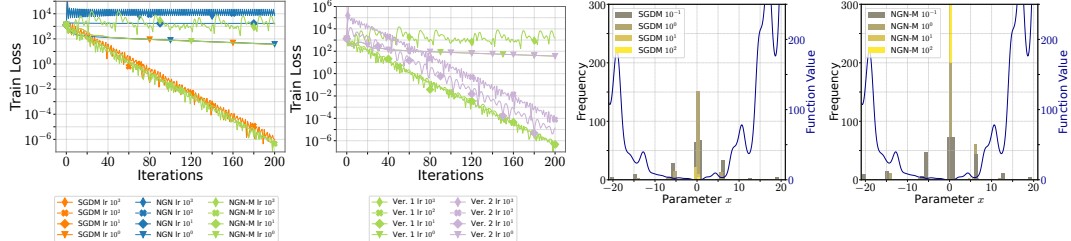

Figure 1: **First:** Comparison of SGDM, NGN, NGN-M for linear regression on normalized Diabetes dataset varying a step-size hyper-parameter. **Second:** Comparison of two options on how momentum can be used in combination with NGN step-size. **Third and fourth:** The distribution of the position of the last iterate after 1000 iterations of SGDM and NGN-M on the function described in Section 3.2.

---

**Algorithm 1** NGN-M

---

1: **Input:** $x^{-1} = x^0 \in \mathbb{R}^d$, step-size hyper-parameter $c > 0$, momentum parameter $\beta \in [0, 1)$
2: **for** $k = 0, 1, \ldots, K-1$ **do**
3:      Sample batch $S_k \subseteq [n]$ and compute $f_{S_k}$ and $\nabla f_{S_k}(x^k)$
4:      Compute $\gamma_k = \frac{c}{1 + \frac{c}{2 f_{S_k}(x^k)} \|\nabla f_{S_k}(x^k)\|^2}$
5:      Update $x^{k+1} = x^k - (1-\beta)\gamma_k \nabla f_{S_k}(x^k) + \beta(x^k - x^{k-1})$
6: **end for**

---

where $\Sigma_k \in \mathbb{R}^{d \times d}$ is a diagonal matrix that penalizes each parameter with its weight while in vanilla NGN the penalization is the same for all parameters, and $f$ is an objective function we aim to minimize. Performing simple derivations we obtain the following update rule

$$x^{k+1} = x^k - \gamma_k \Sigma_k^{-1} \nabla f(x^k) \quad \text{where} \quad \gamma_k = \frac{c}{1 + \frac{c}{2 f(x^k)} \|\nabla f(x^k)\|^2_{\Sigma_k^{-1}}}. \tag{3}$$

The derivations of the update rule (3) are deferred to Appendix E. By appropriately choosing $\Sigma_k$ we obtain a diagonal version of NGN step-size. Note that by choosing $\Sigma_k$ to be an identity matrix, the step-size $\gamma_k$ in (3) reduces to vanilla NGN step-size.

A possible choice of $\Sigma_k$ is a RMSprop preconditioner. When combined with momentum, this results in a more practical algorithm, which we refer to as NGN-MDv1. (Alg. 2). From an empirical evaluation of NGN-MD in Figure 2, we observe that this choice improves the performance of NGN-M while maintaining robustness to step-size hyper-parameter.

Instead of relying on the minimizing the model in (2) we can follow a more straightforward approach. We can replace the gradient norm in NGN step-size by the $j$-th partial derivative to update the $j$-th parameter. This leads to the update of the form $\Sigma_k^{-1} \nabla f_{S_k}(x^k)$ where $(\Sigma_k)^{-1}_{(j)} = \gamma_k^{(j)} := \frac{c}{1 + \frac{c}{2 f(x^k)} (\nabla_j f(x^k))^2}$. We name the algorithm with this choice of $\Sigma_k$ as NGN-D. We believe that NGN-D is the first algorithm that uses a Polyak-type step-size per coordinate while at least achieving the standard $\mathcal{O}(1/\sqrt{K})$ convergence rate under smoothness and bounded noise variance assumptions (see Theorem 2). Even though the convergence guarantees of NGN-D are interesting on its own, we defer the detailed NGN-D description and its convergence to Appendix C as the resilience of NGN-M to the step-size hyper-parameter tuning is the main focus of the paper. A more detailed discussion on the two versions of NGN-MD algorithms is deferred to Appendix E.1 together with the computation cost of their step in Appendix E.2.

## 4 THEORETICAL ANALYSIS OF NGN-M

### 4.1 PROBLEM FORMULATION AND NOTATION

We consider the classic Empirical Risk Minimization (ERM) problem that typically appears when training machine learning models, namely,

$$\min_{x \in \mathbb{R}^d} \left[ f(x) := \frac{1}{n} \sum_{i=1}^n f_i(x) \right], \tag{4}$$

---

**Algorithm 2** NGN-MD

---

1: **Input:** $x^0 \in \mathbb{R}^d$, step-size hyper-parameter $c > 0$, momentum parameters $\beta_1, \beta_2 \in [0, 1)$, stabilization parameter $\varepsilon > 0$
2: **for** $k = 0, 1, \ldots, K - 1$ **do**
3:     Sample batch $S_k \subseteq [n]$ and compute $f_{S_k}$ and $\nabla f_{S_k}(x^k)$
4:     Compute $v^k = \beta_2 v^{k-1} + (1 - \beta_2)(\nabla f_{S_k}(x^k) \odot \nabla f_{S_k}(x^k))$
5:     Compute $\mathbf{D}_k = \mathrm{diag}(\varepsilon \mathbf{I} + \sqrt{v^k/(1 - \beta_2^k)})$
6:     Compute $\gamma_k = \frac{c}{1 + \frac{c}{2 f_{S_k}(x^k)} \|\nabla f_{S_k}(x^k)\|^2_{\mathbf{D}_k^{-1}}}$                    only for NGN-MDv1
7:     Compute $\mathbf{\Sigma}_k^{-1} = \gamma_k \mathbf{D}_k^{-1}$                    for NGN-MDv1
8:     Compute $\mathbf{\Sigma}_k^{-1} = \mathrm{diag}\left( \frac{c/(\mathbf{D}_k)_{(j)}}{1 + \frac{c}{2 f_{S_k}(x^k) \cdot (\mathbf{D}_k)_{(j)}}(\nabla_j f_{S_k}(x^k))^2} \right)$                    for NGN-MDv2
9:     Update $x^{k+1} = x^k - (1 - \beta_1)\mathbf{\Sigma}_k^{-1}\nabla f_{S_k}(x^k) + \beta_1(x^k - x^{k-1})$
10: **end for**

---

where $x$ are the parameters of a model we aim to train, $n$ is the number of data points in the dataset, $d$ is the number of parameters, and $f_i$ represents the loss associated with the $i$-th data point/batch. We assume that each $f_i$ is differentiable and non-negative[3] and that the global optimal value is bounded, i.e. $f^* = \arg\min_x f(x) \in \mathbb{R}$. Moreover, we assume that we have access to mini-batch stochastic losses $f_S$ during training such that $f_S^* := \arg\min_x f_S(x) < \infty$ for any $S \subseteq [n]$ picked uniformly at random.

Next, we provide the definitions that are frequently used in the analysis.

**Definition 1.** *The function $\phi \colon \mathbb{R}^d \to \mathbb{R}$ is convex if for all $x, y \in \mathbb{R}^d$ we have*

$$\langle \nabla f(x), y - x \rangle \geq f(x) - f(y). \tag{5}$$

**Assumption 1.** *We assume that the interpolation $\sigma_{\mathrm{int}}^2 := \mathbb{E}_S[f^* - f_S^*]$ and positive $\sigma_{\mathrm{pos}}^2 := \mathbb{E}_S[f_S^*]$ errors are bounded by real numbers $\sigma_{\mathrm{int}}^2$ and $\sigma_{\mathrm{pos}}^2$ correspondingly.*

Convexity and the aforementioned noise structure are commonly used assumptions in the context of Polyak-like step-sizes (Loizou et al., 2021; Orvieto et al., 2022; Jiang & Stich, 2024; Orvieto & Xiao, 2024; Oikonomou & Loizou, 2024; Schaipp et al., 2024). We say that the interpolation holds if $\sigma_{\mathrm{int}}^2 = 0$.

### 4.2 CONVERGENCE GUARANTEES

**Theorem 1.** *Assume that each $f_i$ is convex and $L$-smooth and that Assumption 1 holds. Let the step-size hyper-parameter $c > 0$ and the momentum parameter $\beta = \frac{\lambda}{1+\lambda}$ be constants where $\lambda \leq \min\{cL, 0.5(1 + cL)^{-1}(1 + 2cL)^{-1}\}$. Then the iterates of NGN-M (Alg. 1) satisfy*

$$\mathbb{E}\left[f(\overline{x}^{K-1}) - f(x^*)\right] \leq \frac{\|x^0 - x^*\|^2}{\rho K} + \frac{8c^2 L}{\rho}\sigma_{\mathrm{int}}^2 + \frac{1}{\rho}\frac{2c^2 L}{1 + cL}\max\left\{\frac{2cL-1}{2cL+1}, 0\right\}\sigma_{\mathrm{pos}}^2, \tag{6}$$

*where $\overline{x}^{K-1}$ is chosen uniformly at random from $\{x^0, \ldots, x^{K-1}\}$, $\rho = \frac{c}{(1+cL)(1+2cL)}$. Moreover, if we set $c = \mathcal{O}(1/\sqrt{K})$ then we obtain $\mathbb{E}\left[f(\overline{x}^{K-1}) - f(x^*)\right] \leq \mathcal{O}(1/\sqrt{K})$.*

In more detail, we observe that $(i)$ NGN-M converges with the same rate as SGDM (Garrigos & Gower, 2023) in the convex setting. The analysis is performed under standard smoothness and convexity assumptions. In contrast, convergence guarantees in previous works that combine SPS and momentum require strong assumptions such as bounded gradients and interpolation, or bounded domain. $(ii)$ NGN-M converges to the exact solution while algorithms such as MomSPS and ALR-SMAG were shown to converge up to a non-vanishing neighborhood of the solution only[4]. Notably, the non-vanishing neighborhood disappears when the problem satisfies interpolation. We refer to Table 1 for more details and exact rates. $(iii)$ The step-size hyper-parameter $c$ is not constrained to

---

[3]Common losses, e.g. cross-entropy, satisfy this condition.
[4]In fact, this is an inherited property of SPS analysis from (Loizou et al., 2021).

be on the order of $\mathcal{O}(1/L)$, as is commonly required in the analysis of gradient-based algorithms.$(iv)$ Finally, in the special case where momentum is absent, i.e. $\lambda = 0$, there are no requirements on the step-size hyper-parameter $c$, similarly to the results by Orvieto & Xiao (2024), which shows the tightness of our analysis.

The convergence theory and detailed algorithm description of NGN-D are deferred to Appendix C. We highlight that to the best of our knowledge, NGN-D is the first algorithm that uses a diagonal Polyak-type step-size and attains the standard convergence rate for general non-convex functions under the Polyak-Łojasiewicz condition.

### 4.3 Key Ingredients of the Proof

We discuss the key steps of the proof to highlight the main challenges in the analysis.

First, we make use of the Iterative Moving Average (IMA) formulation of momentum (Sebbouh et al., 2021). Specifically, we define a sequence of virtual iterates $\{z^k\}$ whose update rule is of the form

$$z^{k+1} = x^k - \gamma_k \nabla f_{S_k}(x^k), \quad x^{k+1} = \frac{\lambda}{1+\lambda}x^k + \frac{1}{1+\lambda}z^{k+1}, \quad \text{where } z^0 := x^0 \text{ and } \beta = \frac{\lambda}{1+\lambda}. \quad (7)$$

Next, one of the key technical strategies we follow is splitting the step-size $\gamma_k$ into two parts: a non-adaptive term $\rho = \frac{c}{(1+cL)(1+2cL)} = \mathcal{O}(c)$ and an adaptive term $\widetilde{\gamma}_k \leq \frac{3c^2L}{1+2cL} = \mathcal{O}(c^2)$. In the analysis, this decomposition of the step-size $\gamma_k$ enables us to regulate the balance between the descent term, which drives improvement in the objective, and the error term, which reflects possible inaccuracies. More precisely, the descent term is weighted by $c$ while the error term proportional to $\sigma_{\mathrm{int}}^2$ is weighted by $c^2$, which suggests that $c$ has to be chosen to trade off the two terms to lead to the exact convergence similarly to the standard analysis of SGD (Garrigos & Gower, 2023). In contrast, MomSPS and Momo algorithms achieve the exact convergence only under the interpolation regime.

## 5 Experiments

We now turn to the empirical evaluation of the proposed algorithms against several benchmarks. The detailed experiment setup, including the choice of hyper-parameters as well as additional experimental results and details, can be found in Appendix G. The best performance of algorithms is reported in Tables 3 (momentum-based algorithms), 4 (algorithms with momentum and diagonal step-size), and 5 (algorithms with diagonal step-size).

### 5.1 Comparison of Algorithms with Momentum

First, we test the performance of NGN-M against other methods that use momentum such as SGDM, Momo (Schaipp et al., 2024), MomSPS (Oikonomou & Loizou, 2024), and ALR-SMAG (Wang et al., 2023), and NGN (Orvieto & Xiao, 2024) (which already exhibits a high degree of robustness without momentum). The tests include the training of Resnet20 (He et al., 2016) and ViT (Dosovitskiy et al., 2021) on CIFAR10 dataset (Krizhevsky et al., 2009), and Resnet110 on CIFAR100 dataset. All experiments in this section do not use learning rate schedulers or weight decay.

First, from Table 3 we observe that the best performance of NGN-M matches the results of other algorithms (the interval of one standard deviation of validation score of NGN-M always intersects with the interval of the best algorithm). This demonstrates that tuned NGN-M exhibits competitive performance across all settings we tested. Importantly, NGN-M demonstrates significantly greater robustness to the choice of the step-size hyper-parameter. Indeed Figure 2 shows that the range of step-size hyper-parameters that allows NGN-M to perform optimally is much wider. We can for instance use step-sizes that are 1-2 orders in magnitude larger than the optimal one without a significant drop in the performance. This is particularly evident during the training of Resnet20 and ViT. Besides, we clearly observe that momentum consistently improves the stability of NGN across all settings. We refer to Appendix G.2 for the train loss stability and to Appendix G.5 for additional comparison against Lion, Adabound, and Adabelief, and to Appendix G.10 for the results in training NLP models.

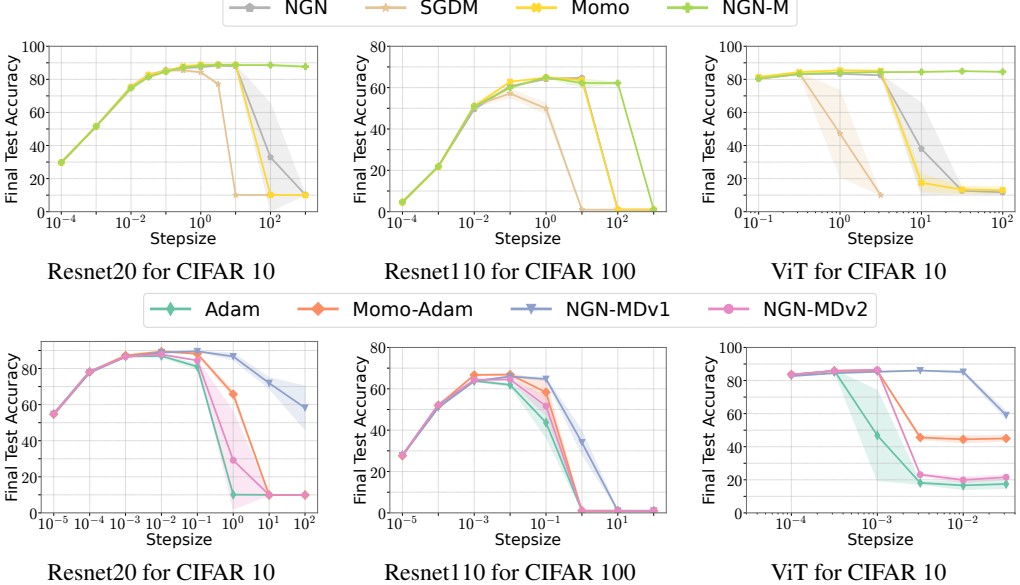

Figure 2: Stability performance of algorithms varying step-size hyper-parameter ($c$ for NGN-M, NGN-MDv1 and NGN-MDv2, $\alpha_0$ for Momo and Momo-Adam, and step-size for SGDM and Adam). For NGN-M and NGN-MDv1, we observe that the range of the step-size hyper-parameters that provide competitive performance is wider than that for other algorithms. We refer to Figures 7 to 9 and 11 to 13 for train loss stability and for the results on additional workloads.

## 5.2 COMPARISON OF ALGORITHMS WITH MOMENTUM AND DIAGONAL STEP-SIZE

Next, we test the performance of NGN-MDv1 and NGN-MDv2 against other methods that use both momentum and diagonal step-size such as Adam and Momo-Adam (Schaipp et al., 2024). We use the same set of workloads as in Section 5.1. All experiments in this section do not use any learning rate schedulers or weight decay.

Again, in Table 4 we observe that NGN-MDv1 always matches the performance of the best optimizer while NGN-MDv2 is slightly worse in the training of both Resnet models and LSTM models, but better in the training of ViT. On top of this, NGN-MDv1 outperforms other competitors in terms of stability w.r.t. step-size hyper-parameter tuning. The results in Figure 2 showcase that, for NGN-MDv1, we can use a step-size hyper-parameter 1-2 orders of magnitude larger without noticeably hurting the performance. In contrast, competing optimizers do not exhibit a competitive performance for large step-size hyper-parameters. We refer to Appendix G.3 for the train loss stability results and to Appendix G.5 for the additional comparison against Lion, Adabelief, and Adabound.

## 5.3 VISION EXPERIMENTS ON IMAGENET

Having observed promising results on workloads of small and medium size, we switch to larger tasks and datasets. We first train a ResNet18 on ImageNet1k (Deng et al., 2009). This represents the first task in which we pair our proposed algorithms with a learning rate schedule. As illustrated in Figure 3 and Table 3, NGN-M achieves the highest validation accuracy, while exhibiting higher robustness across larger step-sizes, improving over both NGN and Momo. Among adaptive methods, NGN-MDv1 compares favorably against Adam and MomoAdam, while once again achieving higher performance on a wider range of learning rates (Table 4). Appendix G.4 reports additional ablations on ImageNet32 and train loss stability results.

We then test the effectiveness of the proposed algorithms on vision transformers (Dosovitskiy et al., 2021). These models are trained for longer horizon compared to convolutional architectures, are notoriously sensitive to initial learning rate, and require adaptive step-sizes. We follow the protocol of Schaipp et al. (2024), which includes cosine annealing, but without any weight decay regularization. As highlighted in Figure 3 and Table 4, NGN-MDv1 achieves the highest validation accuracy across adaptive methods. Moreover, at a larger learning rate, Adam diverges, whereas both MomoAdam NGN-MDv1 maintain more stable training dynamics.

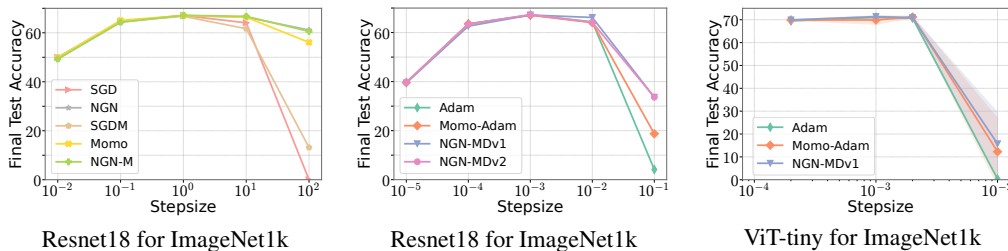

Figure 3: Stability performance on ImageNet1k varying the step-size hyper-parameter. NGN-M and algnameNGN-MDv1 achieve higher accuracy for a wider range of the step-size hyper-parameters. We refer to Figure 10 for results on train loss stability and additional results on ImageNet32.

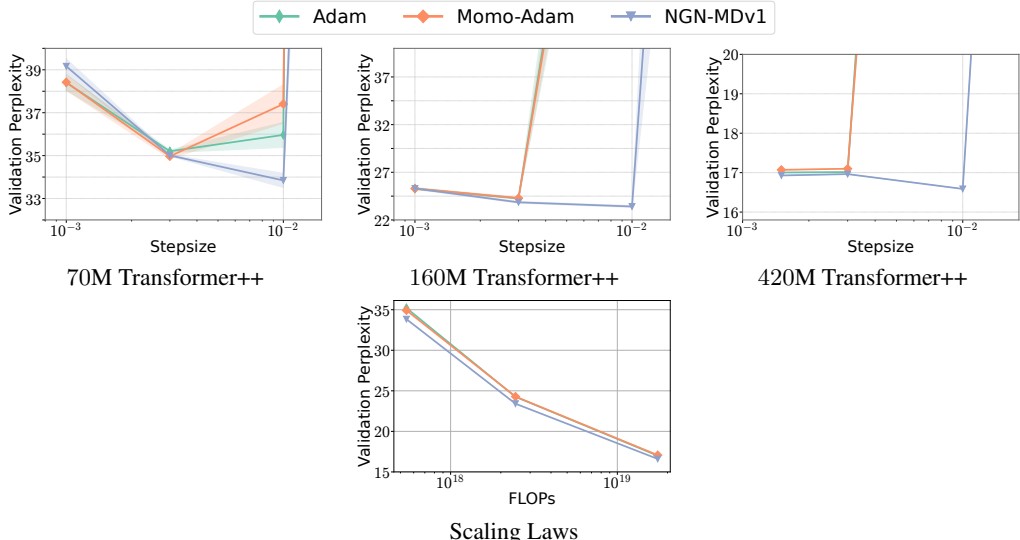

Figure 4: Language Modeling on SlimPajama. **First row:** stability comparison with respect to the step-size hyper-parameter across different model sizes and optimizers. At all model capacities, NGN-MDv1 achieves the lowest perplexity, showing better stability and improved performance at larger learning rates. **Second row:** the scaling laws for the three algorithms, highlighting the effectiveness of NGN-MDv1 over Adam and Momo-Adam across all tested scales.

## 5.4 LANGUAGE MODELING

Pre-training Large Language Models represents a challenging optimization task. To achieve competitive performance, optimizers with adaptive step-size are needed, and preventing instabilities in low-precision training often requires careful hyper-parameter tuning.

To evaluate the capability of NGN-MDv1 in this setting, we train decoder-only transformers (Radford et al., 2019) with 70M, 160M, and 420M parameters around Chinchilla optimum (Hoffmann et al., 2022) on SlimPajama-627B (Soboleva et al., 2023). For each model, we retune the learning rate, using 3 seeds for the first two models and 1 seed for the 420M. Appendix G provides additional details about the training and tokenization pipeline.

Figure 4 and Table 4 report the final validation perplexity of the three medium-scale Language Models, as well as scaling laws for different optimizers. We note that both NGN-MDv1 and Momo-Adam match the performance of Adam at its optimal learning rate of $3 \cdot 10^{-3}$. However, at larger step-size $10^{-2}$, Momo and Adam face unrecoverable instabilities, whereas, as reported in Figure 21, NGN-MDv1 remains stable throughout training. This phenomenon is consistent across all scales we tested, suggesting that the optimal learning rate of NGN-MDv1 is shifted towards larger values, but also that the algorithm is less sensible to such hyper-parameter.

In addition to the findings presented in this section, Appendix F discusses how to introduce weight decay in NGN-MDv1, and reports additional ablations on its role in this training task.

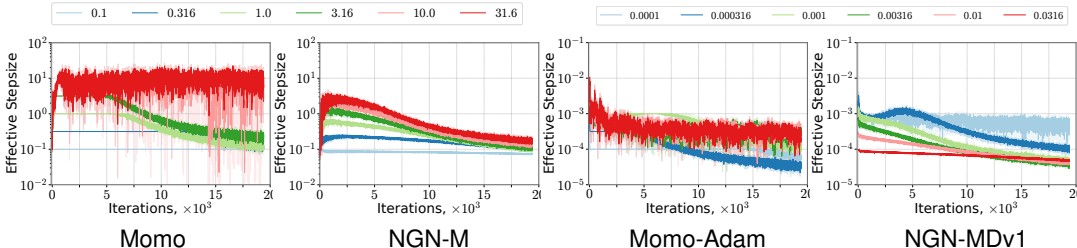

Figure 5: The step-size of Momo, NGN-M (**two left**), Momo-Adam and NGN-MDv1 (**two right**) during the training of ViT on CIFAR10. We demonstrate the step-sizes $\tau_k$ for Momo and Momo-Adam and $\gamma_k$ for NGN-M and NGN-MDv1 varying step-size parameters $\alpha_0$ and $c$ correspondingly. We refer to Figures 14 and 15 for the results in training Resnet20.

## 5.5 CONVERGENCE TO FLATTER MINIMA

Following the discussion in Section 3.2, we conduct a similar evaluation when training a Resnet20 network; see results in Figure 16. We use a code base from Golmant et al. (2018). In particular, we evaluate the test and training loss at the final point reached by NGN-M and SGDM along the eigenvectors corresponding to the first two largest by-magnitude eigenvalues. Increasing the step-size hyper-parameter of NGN-M leads to convergence to flatter minima at the same test and training loss levels. This fact explains why a larger step-size hyper-parameter does not hurt training. Conversely, SGDM diverges for a large step-size value. We additionally demonstrate the evolution of the spectrum during the training with NGN-M and SGDM in Figures 19 and 20. We refer to Appendix G.8 for a more detailed description of the observed phenomenon.

## 5.6 EFFECTIVE STEP-SIZE OF NGN-M AND NGN-MDv1

The first observation from the results in Figure 5 is that the effective step-size of NGN-M and NGN-MDv1 is always adaptive: if the step-size hyper-parameter $c$ is large enough the effective step-size sharply increases in the beginning up to a peak, and then it gradually decreases till the end of the training. From this perspective, NGN-M and NGN-MDv1 step-sizes are close to annealing step-size schedulers widely used in practice. In contrast, the effective step-size of Momo and Momo-Adam is not adaptive for sufficiently large step-size hyper-parameter $\alpha_0$ during the initial part or all of the training. In other words, these algorithms reduce to SGDM and Adam which is one of the reasons for the reduced resilience property of Momo and Momo-Adam in comparison with NGN-M and NGN-MDv1. The effective step-sizes in training Resnet20 are provided Figures 14 and 15.

## 6 CONCLUSION AND FUTURE WORK

This work introduced several novel adaptations of the NGN step-size method, incorporating support for momentum and/or diagonal step-size. We provided a theoretical analysis of the convergence rates for these algorithms and conducted an extensive empirical evaluation of their performance. The experimental results show that combining momentum with the NGN step-size yields high robustness to step-size hyper-parameter choices and performs competitively with state-of-the-art algorithms across various settings.

Given the significant complexity of the task, we defer the theoretical explanation of the step-size resilience properties of NGN-M and analysis in the non-convex setting to future work. Furthermore, while the two proposed methods for incorporating weight decay into NGN-MDv1 outperform AdamW and Momo-AdamW in training language models, they still exhibit some sensitivity to the step-size hyper-parameter. This may, in part, be due to the limited understanding of the expected effects of the weight decay technique, a topic that requires further investigation. Finally, one might question the reasons behind the improvements of NGN-MDv1 over Adam, which could stem from the sub-optimal use of momentum in Adam, a direction deserving further exploration.

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

## A    EQUIVALENT FORMULATIONS OF NGN-M

We remind that the iterates of NGN-M are the following

$$
\begin{aligned}
x^{k+1} &= x^k - (1-\beta)\gamma_k \nabla f_{S_k}(x^k) + \beta(x^k - x^{k-1}) \\
&= x^k - (1-\beta)\frac{c}{1 + \frac{c}{2f_{S_k}(x^k)}\|\nabla f_{S_k}(x^k)\|^2}\nabla f_{S_k}(x^k) + \beta(x^k - x^{k-1}).
\end{aligned}
$$

We can rewrite the update rule using Iterative-Moving Average (IMA) approach presented in Proposition 1.6, Sebbouh et al. (2021).

**Lemma 1** (Proposition C.8 (Oikonomou & Loizou, 2024), Lemma 7.3 in (Garrigos & Gower, 2023)). *The iterates $\{x^k\}$ generated by NGN-M are equivalent to the sequence $\{x^k\}$ generated by IMA update*

$$
z^{k+1} = x^k - \gamma_k \nabla f_{S_k}(x^k), \quad x^{k+1} = \frac{\lambda}{1+\lambda}x^k + \frac{1}{1+\lambda}z^{k+1}, \tag{8}
$$

*where*

$$
\beta = \frac{\lambda}{1+\lambda}, \qquad z^{k+1} = x^{k+1} + \lambda(x^{k+1} - x^k), \quad \text{and} \quad x^{-1} = z^0 = x^0. \tag{9}
$$

*Proof.* Let the sequences $\{x^k\}$ and $\{z^k\}$ be defined according to Equation (8). Let $\beta$ be defined as $\frac{\lambda}{1+\lambda}$. Then we have

$$
\begin{aligned}
x^{k+1} &= \frac{\lambda}{1+\lambda}x^k + \frac{1}{1+\lambda}z^{k+1} \\
&= \frac{\lambda}{1+\lambda}x^k + \frac{1}{1+\lambda}(z^k - \gamma_k \nabla f_{S_k}(x^k)) \\
&= \frac{\lambda}{1+\lambda}x^k + \frac{1}{1+\lambda}((1+\lambda)x^k - \lambda x^{k-1} - \gamma_k \nabla f_{S_k}(x^k)) \\
&= x^k - \frac{1}{1+\lambda}\gamma_k \nabla f_{S_k}(x^k) + \frac{\lambda}{1+\lambda}(x^k - x^{k-1}).
\end{aligned}
$$

It remains to use Equation (9) as we have $\beta = \frac{\lambda}{1+\lambda}$ and $1 - \beta = 1 - \frac{\lambda}{1+\lambda} = \frac{1}{1+\lambda}$.

$\square$

## B    TECHNICAL LEMMAS AND DEFINITIONS

**Definition 2.** We say that the function $\phi$ admits **L**-smooth with parameters $\mathbf{L} := (L_1, \ldots, L_d), L_j \geq 0 \; \forall j \in [d]$, if the following inequality holds for all $x, h \in \mathbb{R}^d$

$$
\phi(x+h) \leq \phi(x) + \langle \nabla\phi(x), h\rangle + \tfrac{1}{2}h^\top \mathbf{L}h. \tag{10}
$$

**Remark 1.** If we set for all $j \in [d]$ $L_j := L$ then Definition 2 reduces to standard $L$-smoothness.

This assumption is typically used in the context of coordinate adaptive algorithms such as Sign SGD (Bernstein et al., 2018; Safaryan & Richtárik, 2021).

**Definition 3.** The function $\phi\colon \mathbb{R}^d \to \mathbb{R}$ satisfies *PŁ-condition* with constant $\mu > 0$ if for all $x, y \in \mathbb{R}^d$ we have

$$
\|\nabla f(x)\|^2 \geq 2\mu(f(x) - f^*). \tag{11}
$$

**Assumption 2.** *We assume that the coordinate-wise variance of the stochastic estimator is bounded, i.e. for all $x \in \mathbb{R}^d$ and $j \in [d]$ we have*

$$
\mathbb{E}_S\left[|(\nabla_j f_S(x) - \nabla_j f(x)|^2\right] \leq \sigma_j^2. \tag{12}
$$

**Lemma 2** (Lemma 4.9 from (Orvieto & Xiao, 2024)). *Let each $f_i$ be $L$-smooth for all $i$, then the step-size of NGN satisfies*

$$
\gamma_k \in \left[\frac{c}{1+cL}, c\right]. \tag{13}
$$

**Lemma 3** (Lemma 4.2 from (Orvieto & Xiao, 2024)). Let each $f_i$ be $L$-smooth for all $i$, then the iterates of NGN satisfy

$$\gamma_k^2 \|\nabla f_{S_k}(x^k)\|^2 \leq \frac{4cL}{1+2cL} \gamma_k(f_{S_k}(x^k) - f_{S_k}^*) + \frac{2c^2 L}{1+cL} \max\left\{\frac{2cL-1}{2cL+1}, 0\right\} f_{S_k}^*. \tag{14}$$

**Lemma 4** (Gradient Upper Bound). Let $\phi: \mathbb{R}^d \to \mathbb{R}$ satisfy Definition 2. Then, for all $x \in \mathbb{R}^d$ and all $j \in [d]$ we have

$$2L_j(f(x) - f^*) \geq (\nabla_j f(x))^2. \tag{15}$$

*Proof.* From Definition 2 we have

$$f^* = \min_{x \in \mathbb{R}^d} f(x) \leq \min_{h_j \in \mathbb{R}} f(x + h_j e_j) \leq f(x) + \min_{h_j \in \mathbb{R}} \left[\nabla_j f(x) h_j + \frac{L_j}{2} h_j^2\right].$$

Now we can explicitly compute the minimum in the right-hand side. The optimal value is achieved at

$$h_j^* := -\frac{1}{L_j} \nabla_j f(x),$$

therefore,

$$
\begin{aligned}
f^* &\leq f(x) + \nabla_j f(x) h_j^* + \frac{L_j}{2}(h_j^*)^2 \\
&= f(x) - \frac{1}{L_j}(\nabla_j f(x))^2 + \frac{1}{2L_j}(\nabla_j f(x))^2 \\
&= f(x) - \frac{1}{2L_j}(\nabla_j f(x))^2,
\end{aligned}
$$

which equivalent to the statement of the lemma. $\square$

## C  CONVERGENCE OF NGN-D

First, we provide NGN-D pseudocode and the main convergence results.

---

**Algorithm 3** NGN-D

---

1: **Input:** $x^0 \in \mathbb{R}^d$, step-size parameter $c > 0$
2: **for** $k = 0, 1, \ldots, K-1$ **do**
3:      Sample batch $S_k \subseteq [n]$ and compute $f_{S_k}$ and $\nabla f_{S_k}(x^k)$
4:      Compute $\gamma_k^{(j)} = \frac{c}{1 + \frac{c}{2f_{S_k}(x^k)}(\nabla_j f_{S_k}(x^k))^2}$
5:      Update

$$x_{(j)}^{k+1} = x_{(j)}^k - \gamma_k^{(j)} \nabla_j f_{S_k}(x^k)$$

6: **end for**

---

**Theorem 2.** *Let each $f_i$ satisfies Definition 2. Assume that Assumption 2 holds. Then the iterates of* NGN-D *(Alg. 3) with step-size parameters $\{c_j\}_{j=1}^d$ such that $c_j \leq 1/2L_j$ satisfy*

$$\min_{0 \leq k < K} \mathbb{E}\left[\|\nabla f(x^k)\|^2\right] \leq \frac{12(f(x^0) - f^*)}{c_{\min} K} + \frac{1}{c_{\min}} \sum_{j=1}^d 18 L_j c_j^2 \sigma_j^2, \tag{16}$$

*where $c_{\min} := \min_{j \in [d]} c_j$. Moreover, if $c_j = \mathcal{O}(\varepsilon^2)$ for all $j \in [d]$ then after $K = \mathcal{O}(\varepsilon^{-4})$ we obtain $\min_{0 \leq k < K} \mathbb{E}\left[\|\nabla f(x^k)\|^2\right] \leq \mathcal{O}(\varepsilon^2).$*

NGN-D converges with classic rate $\mathcal{O}(1/\sqrt{K})$ similar to Adagrad (Ward et al., 2020). We highlight that, to the best of our knowledge, NGN-D is the first algorithm that uses diagonal Polyak-type stepsize and converges under standard smoothness and bounded variance assumptions without requirements of bounded gradients and interpolation.

**Theorem 3.** *Let $f$ satisfies PŁ-condition and each $f_i$ satisfies Definition 2. Assume that Assumption 2 holds. Then the iterates of* NGN-D *(Alg. 3) with step-size parameters $\{c_j\}_{j=1}^d$ such that $c_j \leq \min\{1/2L_j, 6/\mu\}$ satisfy*

$$\mathbb{E}\left[f(x^K) - f^*\right] \leq (1 - \mu c_{\min}/6)^K (f(x^0) - f^*) + \frac{9}{\mu c_{\min}} \sum_{j=1}^d L_j c_j^2 \sigma_j^2, \tag{17}$$

*where $c_{\min} := \min_{j \in [d]} c_j$. Moreover, if $c_j = \mathcal{O}(\varepsilon)$ for all $j \in [d]$ then after $K = \max\{\mathcal{O}(\varepsilon^{-1}), \mathcal{O}(1)\} \log \varepsilon^{-1}$ iterations we obtain $\mathbb{E}\left[f(x^K) - f^*\right] \leq \mathcal{O}(\varepsilon)$.*

To the best of our knowledge, this is the first result of the convergence of the Polyak-like step-size algorithm under the PŁ-condition. The convergence guarantees are similar to that of SGD (Garrigos & Gower, 2023).

Now we are ready to derive the step-size bounds.

**Lemma 5** (Step-size Bounds). Let $f_{S_k}(x) \colon \mathbb{R}^d \to \mathbb{R}$ be a stochastic loss of batch $S_k$ at iteration $k$. Let $f_{S_k}(x)$ satisfy Definition equation 2. Consider $\gamma_j^k$ as in NGN-D (Algorithm 3), then we have

$$\gamma_j^k \in \left[\frac{c_j}{1 + c_j L_j}, c_j\right]. \tag{18}$$

*Proof.* From Lemma 4 we have $2L_j(f_{S_k}(x^k) - f_{S_k}^*) \geq (\nabla_j f_{S_k}(x^k))^2$. Since we assume that each $f_{S_k}^* \geq 0$, then $2L_j f_{S_k}(x^k) \geq (\nabla_j f_{S_k}(x^k))^2$, or equivalently,

$$0 \leq \frac{(\nabla_j f_{S_k}(x))^2}{2 f_{S_k}(x)} \leq L_j.$$

Therefore, for all $j \in [d]$ we have

$$\gamma_j^k = \frac{c_j}{1 + \frac{c_j}{2 f_{S_k}(x^k)}(\nabla_j f_{S_k}(x^k))^2} \leq \frac{c_j}{1} = c_j,$$

and

$$\gamma_j^k = \frac{c_j}{1 + \frac{c_j}{2 f_{S_k}(x^k)}(\nabla_j f_{S_k}(x^k))^2} \geq \frac{c_j}{1 + c_j L_j},$$

that concludes the proof. $\square$

**Lemma 6** (Fundamental Equality). Consider $\gamma_j^k$ as in NGN-D (Algorithm 3). Then the following equality holds

$$\gamma_j^k (\nabla_j f_{S_k}(x^k))^2 = 2 \left(\frac{c_j - \gamma_j^k}{c_j}\right) f_{S_k}(x^k). \tag{19}$$

*Proof.* From NGN-D (Algorithm 3) we have

$$\left(1 + \frac{c_j}{2 f_{S_k}(x^k)}(\nabla_j f_{S_k}(x^k))^2\right)\gamma_j^k = c_j,$$

which one can rewrite as

$$\frac{c_j}{2 f_{S_k}(x^k)}(\nabla_j f_{S_k}(x^k))^2 \gamma_j^k = c_j - \gamma_j^k.$$

It is left to divide both sides by $\frac{2 f_{S_k}(x^k)}{c_j}$. $\square$

## C.1 Convergence in General Non-convex Setting

**Theorem 2.** *Let each $f_i$ satisfies Definition 2. Assume that Assumption 2 holds. Then the iterates of* NGN-D *(Alg. 3) with step-size parameters $\{c_j\}_{j=1}^d$ such that $c_j \leq 1/2L_j$ satisfy*

$$\min_{0 \leq k < K} \mathbb{E}\left[\|\nabla f(x^k)\|^2\right] \leq \frac{12(f(x^0) - f^*)}{c_{\min}K} + \frac{1}{c_{\min}} \sum_{j=1}^d 18L_j c_j^2 \sigma_j^2, \tag{16}$$

*where $c_{\min} := \min_{j \in [d]} c_j$. Moreover, if $c_j = \mathcal{O}(\varepsilon^2)$ for all $j \in [d]$ then after $K = \mathcal{O}(\varepsilon^{-4})$ we obtain $\min_{0 \leq k < K} \mathbb{E}\left[\|\nabla f(x^k)\|^2\right] \leq \mathcal{O}(\varepsilon^2)$.*

*Proof.* First, we write separable Definition 2

$$
\begin{aligned}
f(x^{k+1}) - f(x^k) &= f\left(x^k - \sum_{j=1}^d \gamma_j^k \nabla_j f_{S_k}(x^k) e_j\right) - f(x^k) \\
&\leq -\sum_{j=1}^d \nabla_j f(x^k) \cdot \gamma_j^k \nabla_j f_{S_k}(x^k) + \frac{1}{2} \sum_{j=1}^d L_j (\gamma_j^k \nabla_j f_{S_k}(x^k))^2 \\
&\leq -\sum_{j=1}^d \nabla_j f(x^k) \cdot \gamma_j^k \nabla_j f_{S_k}(x^k) + \frac{1}{2} \sum_{j=1}^d L_j \sigma_j^2 (\nabla_j f_{S_k}(x^k))^2. \quad (20)
\end{aligned}
$$

Note that both $\gamma_j^k$ and $\nabla_j f_{S_k}(x^k)$ depend on the realization $S_k$, thus we can not directly apply conditional expectation with respect to $x^k$, as in this case we would have to analyze the product $\gamma_j^k \nabla_j f_{S_k}(x^k)$. Given bounds of the step-size $\gamma_j^k$ from Lemma 5, we can write the step-size as follows

$$\gamma_j^k = \frac{c_j}{1 + c_j L_j} + \nu_j^k \frac{c_j^2 L_j}{1 + c_j L_j},$$

where $\nu_j^k \in [0, 1]$ is a random variable. Varying the value of $\nu_j^k$ from 0 to 1 we cover the whole range of $\gamma_j^k$. Thus, we continue as follows

$$
\begin{aligned}
&-\gamma_j^k \nabla_j f(x^k) \nabla_j f_{S_k}(x^k) \\
&= -\frac{c_j}{1 + c_j L_j} \nabla_j f(x^k) \nabla_j f_{S_k}(x^k) - \frac{c_j^2 L_j}{1 + c_j L_j} \nu_j^k \nabla_j f(x^k) \nabla_j f_{S_k}(x^k) \\
&\leq -\frac{c_j}{1 + c_j L_j} \nabla_j f(x^k) \nabla_j f_{S_k}(x^k) + \frac{c_j^2 L_j}{1 + c_j L_j} |\nu_j^k| \cdot |\nabla_j f(x^k) \nabla_j f_{S_k}(x^k)| \\
&\leq -\frac{c_j}{1 + c_j L_j} \nabla_j f(x^k) \nabla_j f_{S_k}(x^k) + \frac{c_j^2 L_j}{1 + c_j L_j} \cdot |\nabla_j f(x^k) \nabla_j f_{S_k}(x^k)|.
\end{aligned}
$$

Now we use the inequality $|ab| \leq \frac{1}{2}a^2 + \frac{1}{2}b^2 + \frac{1}{2}|a - b|^2$, and derive

$$
\begin{aligned}
2\mathbb{E}_k\left[|\nabla_j f(x^k) \nabla_j f_{S_k}(x^k)|\right] &\leq |\nabla_j f(x^k)|^2 + \mathbb{E}_k\left[|\nabla_j f_{S_k}(x^k)|^2\right] + \mathbb{E}_k\left[|\nabla_j f(x^k) - \nabla_j f_{S_k}(x^k)|^2\right] \\
&\leq 2|\nabla_j f(x^k)|^2 + 2\mathbb{E}_k\left[|\nabla_j f(x^k) - \nabla_j f_{S_k}(x^k)|^2\right] \\
&\leq 2|\nabla_j f(x^k)|^2 + 2\sigma_j^2.
\end{aligned}
$$

Therefore, we get

$$
\begin{aligned}
-\mathbb{E}_k\left[\gamma_j^k \nabla_j f(x^k) \nabla_j f_{S_k}(x^k)\right] &\leq -\frac{c_j}{1 + c_j L_j} |\nabla_j f(x^k)|^2 + \frac{c_j^2 L_j}{1 + c_j L_j}\left(|\nabla_j f(x^k)|^2 + \sigma_j^2\right) \\
&= -c_j\left(\frac{1 - c_j L_j}{1 + c_j L_j}\right) |\nabla_j f(x^k)|^2 + \frac{c_j^2 L_j}{1 + c_j L_j} \sigma_j^2. \quad (21)
\end{aligned}
$$

We plug in equation 21 into equation 20 and get

$$
\begin{aligned}
\mathbb{E}_k\left[f(x^{k+1})\right] - f(x^k) &\leq -\sum_{j=1}^{d}\left(\mathbb{E}_k\left[\gamma_j^k \nabla_j f(x^k)\nabla_j f_{S_k}(x^k)\right] + \frac{L_j c_j^2}{2}\mathbb{E}_k\left[|\nabla_j f_{S_k}(x^k)|^2\right]\right) \\
&\leq \sum_{j=1}^{d}\left(\left[-c_j\left(\frac{1-c_jL_j}{1+c_jL_j}\right) + \frac{L_j c_j^2}{2}\right]|\nabla_j f(x^k)|^2 \right. \\
&\qquad\qquad \left. + \left[\frac{c_j^2 L_j}{1+c_jL_j} + \frac{L_j c_j^2}{2}\right]\sigma_j^2\right).
\end{aligned}
$$

If $c_j \leq \frac{1}{2L_j}$, we get

$$
\mathbb{E}_k\left[f(x^{k+1})\right] - f(x^k) \leq \sum_{j=1}^{d}\left(-\frac{c_j}{12}|\nabla_j f(x^k)|^2 + \frac{3L_j c_j^2}{2}\sigma_j^2\right).
$$

$\square$

We continue as follows

$$
\mathbb{E}_k\left[f(x^{k+1})\right] - f(x^k) \leq -\frac{c_{\min}}{12}\|\nabla f(x^k)\|^2 + \sum_{j=1}^{d}\frac{3L_j c_j^2}{2}\sigma_j^2. \tag{22}
$$

Taking full expectation and unrolling the recursion above for all iterations $\{0,\dots,K-1\}$. Thus, we obtain

$$
\min_{0\leq k<K}\mathbb{E}\left[\|\nabla f(x^k)\|^2\right] \leq \frac{1}{K}\sum_{k=0}^{K-1}\mathbb{E}\left[\|\nabla f(x^k)\|^2\right] \leq \frac{12}{c_{\min}K}(f(x^0)-f^*) + \frac{18}{c_{\min}}\sum_{j=1}^{d}L_j c_j^2 \sigma_j^2.
$$

If we choose each $c_j = \frac{c_{0,j}}{\sqrt{K}}$ such that $c_{0,j} \leq \frac{1}{2L_j}$ we ensure that $c_j \leq \frac{1}{2L_j}$ as well. Plugging this step-size into the bound we get

$$
\begin{aligned}
\min_{0\leq k<K}\mathbb{E}\left[\|\nabla f(x^k)\|^2\right] &\leq \frac{12}{\frac{c_{0,\min}}{\sqrt{K}}K}(f(x^0)-f^*) + \frac{18}{\frac{c_{0,\min}}{\sqrt{K}}}\sum_{j=1}^{d}L_j\sigma_j^2\frac{c_{0,j}^2}{K} \\
&\leq \frac{12}{c_{0,\min}\sqrt{K}}(f(x^0)-f^*) + \frac{18}{c_{0,\min}\sqrt{K}}\sum_{j=1}^{d}L_j\sigma_j^2 c_{0,j}^2,
\end{aligned}
$$

where $c_{0,\min} := \min_{j\in[d]} c_{0,j}$. If we choose $K = \mathcal{O}(\varepsilon^{-4})$ we get that

$$
\min_{0\leq k<K}\mathbb{E}\left[\|\nabla f(x^k)\|^2\right] = \mathcal{O}(1/\sqrt{K}) = \mathcal{O}(\varepsilon^2).
$$

## C.2 Convergence under PŁ-condition

**Theorem 3.** *Let $f$ satisfies PŁ-condition and each $f_i$ satisfies Definition 2. Assume that Assumption 2 holds. Then the iterates of NGN-D (Alg. 3) with step-size parameters $\{c_j\}_{j=1}^{d}$ such that $c_j \leq \min\{1/2L_j, 6/\mu\}$ satisfy*

$$
\mathbb{E}\left[f(x^K)-f^*\right] \leq (1-\mu c_{\min}/6)^K(f(x^0)-f^*) + \frac{9}{\mu c_{\min}}\sum_{j=1}^{d}L_j c_j^2\sigma_j^2, \tag{17}
$$

*where $c_{\min} := \min_{j\in[d]} c_j$. Moreover, if $c_j = \mathcal{O}(\varepsilon)$ for all $j \in [d]$ then after $K = \max\{\mathcal{O}(\varepsilon^{-1}), \mathcal{O}(1)\}\log\varepsilon^{-1}$ iterations we obtain $\mathbb{E}\left[f(x^K)-f^*\right] \leq \mathcal{O}(\varepsilon)$.*

*Proof.* We obtain equation 22 and use Definition 3

$$\mathbb{E}_k\left[f(x^{k+1})\right] - f(x^k) \le -\frac{c_{\min}}{12}\|\nabla f(x^k)\|^2 + \sum_{j=1}^{d}\frac{3L_j c_j^2}{2}\sigma_j^2$$

$$\le -\frac{\mu c_{\min}}{6}(f(x^k) - f^*) + \sum_{j=1}^{d}\frac{3L_j c_j^2}{2}\sigma_j^2$$

Subtracting $f^*$ from both sides of the inequality above and taking full expectation we obtain

$$\mathbb{E}\left[f(x^{k+1}) - f^*\right] \le (1 - \mu c_{\min}/6)\mathbb{E}\left[f(x^k) - f^*\right] + \sum_{j=1}^{d}\frac{3L_j c_j^2}{2}\sigma_j^2.$$

Unrolling the recursion above for $\{0,\ldots,K-1\}$ iterations we derive

$$\mathbb{E}\left[f(x^K) - f^*\right] \le (1 - \mu c_{\min}/6)^K(f(x^0) - f^*) + \frac{1}{c_{\min}}\sum_{j=1}^{d}\underbrace{\frac{9L_j\sigma_j^2}{\mu}}_{A_j}c_j^2.$$

Now we follow the proof of Lemma A.3 in Garrigos & Gower (2023). Let us choose $c_j = \min\{1/2L_j, \varepsilon/2dA_j\}$. Together with the choice of $K \ge \max\limits_{j\in[d]}\max\left\{\frac{1}{\varepsilon}\frac{12A_j}{\mu}, \frac{12L_j}{\mu}\right\}\log\frac{2(f(x^0)-f^*)}{\varepsilon}$
we get
$$(1 - \mu c_{\min}/6)^K(f(x^0) - f^*) \le \frac{\varepsilon}{2}.$$

Now we have two cases:

1. $c_{\min}$ does not depend on $\varepsilon$, then we have
$$\frac{1}{c_{\min}}A_j c_j^2 \le \mathcal{O}(\varepsilon^2).$$

2. $c_{\min}$ does depend on $\varepsilon$, i.e. $c_{\min} = \mathcal{O}(\varepsilon)$, then we have
$$\frac{1}{c_{\min}}A_j c_j^2 \le \mathcal{O}(\varepsilon).$$

Therefore, combining all together we get
$$\mathbb{E}\left[f(x^K) - f^*\right] \le \mathcal{O}(\varepsilon)$$

after $K \ge \max\limits_{j\in[d]}\max\left\{\frac{1}{\varepsilon}\frac{12A_j}{\mu}, \frac{12L_j}{\mu}\right\}\log\frac{2(f(x^0)-f^*)}{\varepsilon}$ iterations.

□

## D CONVERGENCE OF NGN-M

**Theorem 1.** *Assume that each $f_i$ is convex and $L$-smooth and that Assumption 1 holds. Let the step-size hyper-parameter $c > 0$ and the momentum parameter $\beta = \frac{\lambda}{1+\lambda}$ be constants where $\lambda \le \min\{cL, 0.5(1 + cL)^{-1}(1 + 2cL)^{-1}\}$. Then the iterates of NGN-M (Alg. 1) satisfy*

$$\mathbb{E}\left[f(\overline{x}^{K-1}) - f(x^*)\right] \le \frac{\|x^0 - x^*\|^2}{\rho K} + \frac{8c^2 L}{\rho}\sigma_{\mathrm{int}}^2 + \frac{1}{\rho}\frac{2c^2 L}{1+cL}\max\left\{\frac{2cL-1}{2cL+1}, 0\right\}\sigma_{\mathrm{pos}}^2, \quad (6)$$

*where $\overline{x}^{K-1}$ is chosen uniformly at random from $\{x^0,\ldots,x^{K-1}\}$, $\rho = \frac{c}{(1+cL)(1+2cL)}$. Moreover, if we set $c = \mathcal{O}(1/\sqrt{K})$ then we obtain $\mathbb{E}\left[f(\overline{x}^{K-1}) - f(x^*)\right] \le \mathcal{O}(1/\sqrt{K})$.*

**Remark 2.** In fact, if $\lambda \le \frac{1}{(1+cL)(1+2cL)}$, then it implies that $\lambda \le \frac{1}{cL}$ because $\frac{1}{x} > \frac{1}{(1+x)(1+2x)}$ for any $x > 0$.

*Proof.* To prove the convergence of NGN-M we consider IMA formulation Equation (8):

$$x^{-1} = z^0 = x^0, \quad z^{k+1} = x^k - \gamma_k \nabla f_{S_k}(x^k), \quad x^{k+1} = \frac{\lambda}{1+\lambda} x^k + \frac{1}{1+\lambda} z^{k+1},$$

where $\beta = \frac{\lambda}{1+\lambda}, z^{k+1} = x^{k+1} + \lambda(x^{k+1} - x^k)$.

At iteration $k = 0$ we have

$$z^1 = z^0 - \gamma_0 \nabla f_{S_0}(x^0) = x^0 - \gamma_0 \nabla f_{S_0}(x^0).$$

Therefore, we get

$$\|z^1 - x^*\|^2 = \|z^0 - x^*\|^2 - 2\gamma_0 \langle \nabla f_{S_0}(x^0), z^0 - x^* \rangle + \gamma_0^2 \|\nabla f_{S_0}(x^0)\|^2$$

$$\overset{\text{Lem.3}}{\leq} \|z^0 - x^*\|^2 - 2\gamma_0 \langle \nabla f_{S_0}(x^0), x^0 - x^* \rangle + \frac{4cL}{1+2cL} \gamma_0 (f_{S_0}(x^0) - f_{S_0}^*)$$

$$+ \frac{2c^2 L}{1+cL} \max\left\{ \frac{2cL-1}{2cL+1}, 0 \right\} f_{S_0}^*. \tag{23}$$

Let $\gamma_0 = \rho + \widetilde{\gamma}_0$ where $\rho = \frac{c}{(1+cL)(1+2cL)}$. Then we have

$$\widetilde{\gamma}_0 = \gamma_0 - \rho$$

$$\overset{\text{Lem.2}}{\leq} c - \frac{c}{(1+cL)(1+2cL)}$$

$$= c\frac{1 + 3cL + 2c^2 L^2 - 1}{(1+cL)(1+2cL)}$$

$$= c^2 L\frac{3 + 3cL}{(1+cL)(1+2cL)}$$

$$= \frac{3c^2 L}{1+2cL}.$$

Using the above we continue from (23)

$$\|z^1 - x^*\|^2 \overset{\text{conv.}}{\leq} \|z^0 - x^*\|^2 - 2\gamma_0(f_{S_0}(x^0) - f_{S_0}(x^*)) + \frac{4cL}{1+2cL} \gamma_0(f_{S_0}(x^0) - f_{S_0}^*)$$

$$+ \frac{2c^2 L}{1+cL} \max\left\{ \frac{2cL-1}{2cL+1}, 0 \right\} f_{S_0}^*$$

$$\leq \|z^0 - x^*\|^2 - 2\rho(f_{S_0}(x^0) - f_{S_0}(x^*)) - 2\widetilde{\gamma}_0(f_{S_0}(x^0) - f_{S_0}^*) + 2\widetilde{\gamma}_0(f_{S_0}(x^*) - f_{S_0}^*)$$

$$+ \frac{4cL}{1+2cL} \gamma_0(f_{S_0}(x^0) - f_{S_0}^*) + \frac{2c^2 L}{1+cL} \max\left\{ \frac{2cL-1}{2cL+1}, 0 \right\} f_{S_0}^*$$

$$= \|z^0 - x^*\|^2 - 2\rho(f_{S_0}(x^0) - f_{S_0}(x^*)) - 2\left( \gamma_0 - \rho - \frac{2cL}{1+2cL} \gamma_0 \right)(f_{S_0}(x^0) - f_{S_0}^*)$$

$$+ 2\widetilde{\gamma}_0(f_{S_0}(x^*) - f_{S_0}^*) + \frac{2c^2 L}{1+cL} \max\left\{ \frac{2cL-1}{2cL+1}, 0 \right\} f_{S_0}^*. \tag{24}$$

Here we have

$$\gamma_0 - \rho - \frac{2cL}{1+2cL} \gamma_0 = \frac{1}{1+2cL} \gamma_0 - \rho$$

$$= \frac{1}{1+2cL} \gamma_0 - \frac{c}{(1+cL)(1+2cL)}$$

$$\overset{\text{Lem.2}}{\geq} \frac{1}{1+2cL} \frac{c}{1+cL} - \frac{c}{(1+cL)(1+2cL)}$$

$$= 0,$$

$\widetilde{\gamma}_0 \leq \frac{3c^2 L}{1+2cL}$, and $f_{S_0}(x^0) - f_{S_0}^* \geq 0$. Hence, we get

$$\|z^1 - x^*\|^2 \leq \|z^0 - x^*\|^2 - 2\rho(f_{S_0}(x^0) - f_{S_0}(x^*)) + \frac{6c^2 L}{1+2cL}(f_{S_0}(x^*) - f_{S_0}^*)$$

$$+ \frac{2c^2 L}{1+cL} \max\left\{ \frac{2cL-1}{2cL+1}, 0 \right\} f_{S_0}^*.$$

Rearranging terms and taking expectation we get

$$2\rho\mathbb{E}\left[f(x^0) - f(x^*)\right] \le \mathbb{E}\left[\|z^1 - x^*\|^2\right] - \|z^0 - x^*\|^2 + \frac{6c^2L}{1 + 2cL}\sigma_{\text{int}}^2$$

$$+ \frac{2c^2L}{1 + cL}\max\left\{\frac{2cL - 1}{2cL + 1}, 0\right\}\sigma_{\text{pos}}^2. \tag{25}$$

Next, for $k > 0$ we can use the relation $z^k = x^k + \lambda(x^k - x^{k-1})$. We expand $\|z^{k+1} - x^*\|^2$

$$\begin{aligned}
\|z^{k+1} - x^*\|^2 &= \|x^k - x^*\|^2 - 2\gamma_k\langle\nabla f_{S_k}(x^k), x^k - x^*\rangle + \gamma_k^2\|\nabla f_{S_k}(x^k)\|^2 \\
&\stackrel{\text{Lem.}1}{=} \|x^k - x^*\|^2 - 2\gamma_k\langle\nabla f_{S_k}(x^k), x^k - x^*\rangle - 2\gamma_k\lambda\langle\nabla f_{S_k}(x^k), x^k - x^{k-1}\rangle \\
&\quad + \gamma_k^2\|\nabla f_{S_k}(x^k)\|^2 \\
&\stackrel{\text{conv.}}{\le} \|x^k - x^*\|^2 - 2\gamma_k(f_{S_k}(x^k) - f_{S_k}(x^*)) - 2\gamma_k\lambda(f_{S_k}(x^k) - f_{S_k}(x^{k-1})) \\
&\quad + \gamma_k^2\|\nabla f_{S_k}(x^k)\|^2 \\
&\stackrel{\text{Lem.}3}{\le} \|x^k - x^*\|^2 - 2\gamma_k(f_{S_k}(x^k) - f_{S_k}(x^*)) - 2\gamma_k\lambda(f_{S_k}(x^k) - f_{S_k}(x^{k-1})) \\
&\quad + \frac{4cL}{1 + 2cL}\gamma_k(f_{S_k}(x^k) - f_{S_k}^*) + \frac{2c^2L}{1 + cL}\max\left\{\frac{2cL - 1}{2cL + 1}, 0\right\}f_{S_k}^*.
\end{aligned}$$

Let $\gamma_k = \rho + \widetilde{\gamma}_k$, where $\rho, \widetilde{\gamma}_k \ge 0$, and $\rho$ is a constant step-size independent of $S_k$ which will be defined later. Therefore, we have

$$\begin{aligned}
\|z^{k+1} - x^*\|^2 &\le \|x^k - x^*\|^2 - 2\rho(f_{S_k}(x^k) - f_{S_k}(x^*)) - 2\widetilde{\gamma}_k(f_{S_k}(x^k) - f_{S_k}(x^*)) \\
&\quad - 2\gamma_k\lambda_t(f_{S_k}(x^k) - f_{S_k}^*) + 2\gamma_k\lambda(f_{S_k}(x^{k-1}) - f_{S_k}^*) \\
&\quad + \frac{4cL}{1 + 2cL}\gamma_k(f_{S_k}(x^k) - f_{S_k}^*) + \frac{2c^2L}{1 + cL}\max\left\{\frac{2cL - 1}{2cL + 1}, 0\right\}f_{S_k}^* \\
&= \|x^k - x^*\|^2 - 2\rho(f_{S_k}(x^k) - f_{S_k}(x^*)) - 2\widetilde{\gamma}_k(f_{S_k}(x^k) - f_{S_k}^*) + 2\widetilde{\gamma}_k(f_{S_k}(x^*) - f_{S_k}^*) \\
&\quad - 2\gamma_k\lambda(f_{S_k}(x^k) - f_{S_k}^*) + 2\gamma_k\lambda(f_{S_k}(x^{k-1}) - f_{S_k}^*) \\
&\quad + \frac{4cL}{1 + 2cL}\gamma_k(f_{S_k}(x^k) - f_{S_k}^*) + \frac{2c^2L}{1 + cL}\max\left\{\frac{2cL - 1}{2cL + 1}, 0\right\}f_{S_k}^* \\
&= \|x^k - x^*\|^2 - 2\rho(f_{S_k}(x^k) - f_{S_k}(x^*)) - 2\left(\widetilde{\gamma}_k + \gamma_k\lambda - \frac{2cL}{1 + 2cL}\gamma_k\right)(f_{S_k}(x^k) - f_{S_k}^*) \\
&\quad + 2\widetilde{\gamma}_k(f_{S_k}(x^*) - f_{S_k}^*) + 2\gamma_k\lambda(f_{S_k}(x^{k-1}) - f_{S_k}^*) \\
&\quad + \frac{2c^2L}{1 + cL}\max\left\{\frac{2cL - 1}{2cL + 1}, 0\right\}f_{S_k}^*. \tag{26}
\end{aligned}$$

We need to find $\rho$ such that

$$\widetilde{\gamma}_k + \gamma_k\lambda - \frac{2cL}{1 + 2cL}\gamma_k \ge 0$$

Since $\widetilde{\gamma}_k = \gamma_k - \rho$, then we have

$$\gamma_k - \rho + \gamma_k\lambda - \frac{2cL}{1 + 2cL}\gamma_k \ge 0$$

$$\Leftrightarrow \gamma_k\left(1 + \lambda - \frac{2cL}{1 + 2cL}\right) \ge \rho.$$

The inequality above is satisfied if it is satisfied for the lower bound on $\gamma_k$ (which is $c/1+cL$), i.e.

$$\frac{c}{1 + cL}\left(\frac{1}{1 + 2cL} + \lambda\right) \ge \rho.$$

We can take $\rho = \frac{c}{(1+cL)(1+2cL)}$ since $\lambda \geq 0$.

$$\widetilde{\gamma}_k = \gamma_k - \rho$$
$$\leq c - \frac{c}{(1+cL)(1+2cL)}$$
$$= c\frac{1 + 3cL + 2c^2L^2 - 1}{(1+cL)(1+2cL)}$$
$$\leq c^2 L \frac{3 + 3cL}{(1+cL)(1+2cL)}$$
$$= \frac{3c^2 L}{1 + 2cL}.$$

Using the above, we get from (26)

$$\|z^{k+1} - x^*\|^2 \leq \|x^k - x^*\|^2 - 2\rho(f_{S_k}(x^k) - f_{S_k}(x^*)) + 2c\lambda(f_{S_k}(x^{k-1}) - f_{S_k}(x^*))$$
$$+ 2c\lambda(f_{S_k}(x^*) - f_{S_k}^*) + \frac{6c^2 L}{1 + 2cL}(f_{S_k}(x^*) - f_{S_k}^*)$$
$$+ \frac{2c^2 L}{1 + cL} \max\left\{\frac{2cL - 1}{2cL + 1}, 0\right\} f_{S_k}^*.$$

Taking expectations we get

$$\mathbb{E}\left[\|z^{k+1} - x^*\|^2\right] \leq \mathbb{E}\left[\|x^k - x^*\|^2\right] - 2\rho\mathbb{E}\left[f(x^k) - f(x^*)\right] + 2c\lambda\mathbb{E}\left[f(x^{k-1}) - f(x^*)\right]$$
$$+ \left(2c\lambda + \frac{6c^2 L}{1 + 2cL}\right)\sigma_{\text{int}}^2 + \frac{2c^2 L}{1 + cL}\max\left\{\frac{2cL - 1}{2cL + 1}, 0\right\}\sigma_{\text{pos}}^2. \quad (27)$$

Rearranging terms we get

$$2\rho\mathbb{E}\left[f(x^k) - f(x^*)\right] - 2c\lambda\mathbb{E}\left[f(x^{k-1}) - f(x^*)\right] \leq \mathbb{E}\left[\|x^k - x^*\|^2\right] - \mathbb{E}\left[\|z^{k+1} - x^*\|^2\right]$$
$$+ \left(2c\lambda + \frac{6c^2 L}{1 + 2cL}\right)\sigma_{\text{int}}^2$$
$$+ \frac{2c^2 L}{1 + cL}\max\left\{\frac{2cL - 1}{2cL + 1}, 0\right\}\sigma_{\text{pos}}^2. \quad (28)$$

Combining Equation (25) and Equation (28) for iterations $\{1, \ldots, K - 1\}$ we get

$$2\rho\mathbb{E}\left[f(x^0) - f(x^*)\right] + 2\rho\sum_{t=1}^{K-1}\mathbb{E}\left[f(x^k) - f(x^*)\right] - 2c\lambda\sum_{t=1}^{K-1}\mathbb{E}\left[f(x^{k-1}) - f(x^*)\right]$$

$$= 2\rho\sum_{k=0}^{K-1}\mathbb{E}\left[f(x^k) - f(x^*)\right] - 2c\lambda\sum_{k=0}^{T-2}\mathbb{E}\left[f(x^k) - f(x^*)\right]$$

$$\leq (2\rho - 2c\lambda)\sum_{k=0}^{K-1}\mathbb{E}\left[f(x^k) - f(x^*)\right]$$

$$\leq \|z^0 - x^*\|^2 + \frac{6c^2 L}{1 + 2cL}\sigma_{\text{int}}^2 + \frac{2c^2 L}{1 + cL}\max\left\{\frac{2cL - 1}{2cL + 1}, 0\right\}\sigma_{\text{pos}}^2$$

$$+ \left(2c\lambda + \frac{6c^2 L}{1 + 2cL}\right)(K - 1)\sigma_{\text{int}}^2 + (K - 1) \cdot \frac{2c^2 L}{1 + cL}\max\left\{\frac{2cL - 1}{2cL + 1}, 0\right\}\sigma_{\text{pos}}^2$$

$$\leq \|z^0 - x^*\|^2 + \left(2c\lambda + \frac{6c^2 L}{1 + 2cL}\right)K\sigma_{\text{int}}^2 + K \cdot \frac{2c^2 L}{1 + cL}\max\left\{\frac{2cL - 1}{2cL + 1}, 0\right\}\sigma_{\text{pos}}^2. \quad (29)$$

We need to ensure that $\rho - c\lambda > 0$ which is satisfied for $\lambda$ such that

$$\frac{\rho}{2} = \frac{c}{2(1 + cL)(1 + 2cL)} > c\lambda$$
$$\Leftrightarrow 1 > 2\lambda(1 + cL)(1 + 2cL).$$

Note that we also assume that $\lambda \leq cL$. Therefore, from (29) we get

$$
\begin{aligned}
\frac{1}{K} \sum_{k=0}^{K-1} \mathbb{E}\left[f(x^k) - f(x^*)\right] &\leq \frac{\|z^0 - x^*\|^2}{2(\rho - c\lambda)K} + \frac{1}{2(\rho - c\lambda)} \left(2c\lambda + \frac{6c^2 L}{1 + 2cL}\right) \sigma_{\text{int}}^2 \\
&\quad + \frac{1}{2(\rho - c\lambda)} \frac{2c^2 L}{1 + cL} \max\left\{\frac{2cL - 1}{2cL + 1}, 0\right\} \sigma_{\text{pos}}^2 \\
&\leq \frac{\|z^0 - x^*\|^2}{2(\rho - c\lambda)K} + \frac{8c^2 L}{2(\rho - c\lambda)} \sigma_{\text{int}}^2 \\
&\quad + \frac{1}{2(\rho - c\lambda)} \frac{2c^2 L}{1 + cL} \max\left\{\frac{2cL - 1}{2cL + 1}, 0\right\} \sigma_{\text{pos}}^2.
\end{aligned}
\tag{30}
$$

Since $\rho - c\lambda \geq \frac{\rho}{2}$ and setting $\overline{x}^k$ be uniformly at random chosen from $\{x^0, \ldots, x^{K-1}\}$ we get

$$
\mathbb{E}\left[f(\overline{x}^k) - f(x^*)\right] \leq \frac{\|z^0 - x^*\|^2}{\rho K} + \frac{8c^2 L}{\rho} \sigma_{\text{int}}^2 + \frac{1}{\rho} \frac{2c^2 L}{1 + cL} \max\left\{\frac{2cL - 1}{2cL + 1}, 0\right\} \sigma_{\text{pos}}^2.
\tag{31}
$$

Plugging the value of $\rho = \frac{c}{(1+cL)(1+2cL)}$ inside we get

$$
\begin{aligned}
\mathbb{E}\left[f(\overline{x}^k) - f(x^*)\right] &\leq \frac{\|z^0 - x^*\|^2}{cK}(1 + cL)(1 + 2cL) + 8cL(1 + cL)(1 + 2cL)\sigma_{\text{int}}^2 \\
&\quad + 2cL \max\{2cL - 1, 0\} \sigma_{\text{pos}}^2.
\end{aligned}
\tag{32}
$$

Choosing $c = \mathcal{O}(1/\sqrt{K})$ we get

$$
\mathbb{E}\left[f(\overline{x}^k) - f(x^*)\right] \leq \mathcal{O}\left(\frac{\|z^0 - x^*\|^2}{\sqrt{K}} + \frac{\sigma_{\text{int}}^2}{\sqrt{K}} + \frac{\sigma_{\text{pos}}^2}{\sqrt{K}} \max\{2cL - 1, 0\}\right).
\tag{33}
$$

Therefore, if $K \geq \mathcal{O}(\varepsilon^{-2})$ then $\mathbb{E}\left[f(\overline{x}^k) - f(x^*)\right] \leq \mathcal{O}(\varepsilon)$. $\qquad \square$

# E  HOW TO DERIVE DIAGONAL NGN-BASED STEP-SIZE?

Here we provide derivations of how combine NGN and diagonal step-size following Section 3.3 for completeness.

We consider the following model

$$p^k = \arg\min_{p \in \mathbb{R}^d} \left[ f_{\mathbf{\Sigma}_k, c}(x^k + p) := (r(x^k) + \nabla r(x^k)^\top p)^2 + \frac{1}{2c} \|p\|_{\mathbf{\Sigma}_k}^2 \right], \tag{34}$$

where $r(x) = \sqrt{f(x)}$. We compute the gradient of RHS of (34) w.r.t. $p$ and equal it to zero:

$$\nabla_p f_{\mathbf{\Sigma}_k, c}(x^k + p) = 2 \left( r(x^k) + \nabla r(x^k)^\top p \right) \nabla r(x^k) + \frac{1}{c} \mathbf{\Sigma}_k p$$

$$= \left( 2\nabla r(x^k) \nabla r(x^k)^\top + \frac{1}{c} \mathbf{\Sigma}_k \right) p + 2r(x^k) \nabla r(x^k).$$

Therefore, we have

$$p^k = - \left( 2\nabla r(x^k) \nabla r(x^k)^\top + \frac{1}{c} \mathbf{\Sigma}_k \right)^{-1} 2r(x^k) \nabla r(x^k).$$

Using Shermann-Morrison formula $(\mathbf{A} + uv^\top)^{-1} = \mathbf{A}^{-1} - \frac{\mathbf{A}^{-1} uv^\top \mathbf{A}^{-1}}{1 + u^\top \mathbf{A}^{-1} v}$ with $\mathbf{A} = {}^1\!/{}_c \mathbf{\Sigma}_k$ we derive

$$p^k = - \left( c\mathbf{\Sigma}_k^{-1} - \frac{2c^2 \mathbf{\Sigma}_k^{-1} \nabla r(x^k) \nabla r(x^k)^\top \mathbf{\Sigma}_k^{-1}}{1 + 2c\nabla r(x^k)^\top \mathbf{\Sigma}_k^{-1} \nabla r(x^k)} \right) 2r(x^k) \nabla r(x^k)$$

$$= -2cr(x^k) \left( 1 - \frac{2c\nabla r(x^k)^\top \mathbf{\Sigma}_k^{-1} \nabla r(x^k)}{1 + 2c\nabla r(x^k) \mathbf{\Sigma}_k^{-1} \nabla r(x^k)} \right) \mathbf{\Sigma}_k^{-1} \nabla r(x^k)$$

$$= -\frac{2cr(x^k)}{1 + 2c\nabla r(x^k) \mathbf{\Sigma}_k^{-1} \nabla r(x^k)} \mathbf{\Sigma}_k^{-1} \nabla r(x^k).$$

Now we plug-in $r(x^k) = \sqrt{f(x^k)}$ and $\nabla r(x^k) = \frac{1}{2\sqrt{f(x^k)}} \nabla f(x^k)$ and obtain

$$p^k = -\frac{2c\sqrt{f(x^k)}}{1 + 2c\frac{1}{4f(x^k)} \nabla f(x^k)^\top \mathbf{\Sigma}_k^{-1} \nabla f(x^k)} \frac{1}{2\sqrt{f(x^k)}} \mathbf{\Sigma}_k^{-1} \nabla f(x^k)$$

$$= \frac{c}{1 + \frac{c}{2f(x^k)} \|\nabla f(x^k)\|_{\mathbf{\Sigma}_k^{-1}}^2} \mathbf{\Sigma}_k^{-1} \nabla f(x^k).$$

## E.1  DESIGN COMPARISON OF NGN-MDv1 AND NGN-MDv2

The derivations in equation 3 are used to provide an intuition of how one can add a diagonal step-size into NGN by choosing the regularization matrix $\mathbf{\Sigma}_k$. By choosing $\mathbf{\Sigma}_k = \mathbf{D}_k$ we recover the update direction of NGN-MDv1. In this case, we have only one global NGN step-size in front of $\mathbf{D}_k$. The design of NGN-MDv2 follows a more straightforward intuition. In particular, it can be seen as a direct extension of NGN to diagonal case by replacing the squared gradient norm $\|\nabla f_{S_k}(x^k)\|^2$ by the squared partial derivative $(\nabla_j f_{S_k}(x^k))^2$ for each parameter $j \in [d]$.

The main difference in comparison with Adam is the order in which the preconditioning and momentum is applied. In both NGN-MDv1 and NGN-MDv2 we average the preconditioned updates $\mathbf{\Sigma}_k^{-1} \nabla f_{S_k}(x^k)$, i.e. we first apply preconditioning and momentum later. In contrast, in Adam the stochastic gradients are averaged to construct new momentum term, and then the momentum is preconditioned. In other words, the momentum is applied first and then it is followed by preconditioning. We believe this change might be one of the reasons behind the step-size hyper-parameter resilience as well.

In practice, we found out that the tuned performance of NGN-MDv1 is slightly better than that of NGN-MDv2. Moreover, NGN-MDv1 demonstrates higher resilience to the choice of the step-size hyper-parameter than NGN-MDv2.

### E.2 COMPUTATION COST OF NGN-MD

Implementing any version of NGN-MD in practice might be slightly more computationally expensive. However, we highlight that computing a step of NGN-MD does not involve matrix-vector operations since the preconditioner is a diagonal matrix, and the matrix notation is used only for the convenience of presentation. The additional computation cost that we have in NGN-MDv1 is the computation of $\|\nabla f_{S_k}(x^k)\|^2_{\mathbf{D}_k^{-1}}$. This can be done by one pass over the gradient and summing the terms $\frac{1}{(\mathbf{D}_k)_j}(\nabla_j f_{S_k}(x^k))^2$ for $j \in [d]$. This operation does not require additional matrix multiplication and can be computed while updating $\mathbf{D}_k$. The rest of the NGN-MDv1 implementation does not add any significant costly operations in comparison with Adam.

## F HOW TO ADD WEIGHT DECAY TO NGN-MDv1?

Regularization techniques serve a fundamental purpose in minimizing generalization error. Orthogonally to their role for generalization, modern deep learning tasks often benefit from the use of weight decay (Xiao, 2024). Despite its widespread application, the role of weight decay is poorly understood. Andriushchenko et al. (2023) suggested that it might provide implicit regularization by stabilizing the loss in over-parameterized neural networks and helping to balance the bias-variance trade-off that leads to lower training loss in under-parameterized networks. However, even in the case of SGD, there is still uncertainty regarding how the weight decay mechanism should be incorporated, as various implementations may exist (Zhang et al., 2018).

We propose two ways of adding weight decay to NGN-MDv1. The first variant follows the approach of Loshchilov & Hutter (2019), adding decoupled weight decay $\lambda$:

$$x^{k+1} = x^k - \lambda c x^k - (1 - \beta_1)\mathbf{\Sigma}_k^{-1}\nabla f_{S_k}(x^k) + \beta_1(x^k - x^{k-1}). \tag{35}$$

In this update rule, the weight is added separately from the update direction $\mathbf{\Sigma}_k^{-1}\nabla f_{S_k}(x^k)$. We call the resulting algorithm (35) Dec-NGN-MDv1, that stands for decoupled NGN-MDv1.

### F.1 COMBINING NGN-MDv1 AND WEIGHT DECAY REGULARIZATION

We now discuss how to combine NGN-MDv1 and weight decay, following the idea that weight decay should perform weight regularization.

We consider the following model

$$f_{\mathbf{\Sigma}_k,\lambda}(x^k + p) := (r(x^k) + \nabla r(x^k)^\top p)^2 + \frac{1}{2c}\|p\|^2_{\mathbf{\Sigma}_k} + \frac{\lambda}{2}\|x^k + p\|^2_{\mathbf{\Sigma}_k}.$$

By taking the gradient of $f_{\mathbf{\Sigma}_k,\lambda}$ w.r.t. $p$ we get

$$0 = 2(r(x^k) + \nabla r(x^k)^\top p)\nabla r(x^k) + \frac{1}{c}\mathbf{\Sigma}_k p + \lambda\mathbf{\Sigma}_k(x^k + p)$$

$$= \left(2\nabla r(x^k)\nabla r(x^k)^\top + \frac{1}{c}\mathbf{\Sigma}_k + \lambda\mathbf{\Sigma}_k\right)p + 2r(x^k)\nabla r(x^k) + \lambda\mathbf{\Sigma}_k x^k.$$

Therefore, we get

$$p^k = -\left(2\nabla r(x^k)\nabla r(x^k)^\top + \frac{1}{c}\mathbf{\Sigma}_k + \lambda\mathbf{\Sigma}_k\right)^{-1}(2r(x^k)\nabla r(x^k) + \lambda\mathbf{\Sigma}_k x^k).$$

Using Sherman-Morrison formula $(\mathbf{A} + uv^\top)^{-1} = \mathbf{A}^{-1} - \frac{\mathbf{A}^{-1}uv^\top\mathbf{A}^{-1}}{1+u^\top\mathbf{A}^{-1}v}$ with $\mathbf{A} = (\lambda + 1/c)\mathbf{\Sigma}_k$ and $u = v = \sqrt{2}\nabla r(x^k)$ we get that

$$\left(2\nabla r(x^k)\nabla r(x^k)^\top + \frac{1}{c}\mathbf{\Sigma}_k + \lambda\mathbf{\Sigma}_k\right)^{-1}$$

$$= \frac{c}{1+\lambda c}\mathbf{\Sigma}_k^{-1} - \frac{\frac{2c^2}{(1+\lambda c)^2}\mathbf{\Sigma}_k^{-1}\nabla r(x^k)\nabla r(x^k)^\top\mathbf{\Sigma}_k^{-1}}{1 + \frac{2c}{1+\lambda c}\nabla r(x^k)\mathbf{\Sigma}_k^{-1}\nabla r(x^k)}.$$

---

**Algorithm 4** NGN-MDv1W

---

1: **Input:** $x^0 \in \mathbb{R}^d$, step-size parameter $c > 0$, momentum parameters $\beta_1, \beta_2 \in [0, 1)$, weight decay parameter $\lambda \geq 0$, stabilization parameter $\varepsilon > 0$

2: **for** $k = 0, 1, \ldots, K - 1$ **do**

3:      Sample batch $S_k \subseteq [n]$ and compute $f_{S_k}$ and $\nabla f_{S_k}(x^k)$

4:      Compute $v^k = \beta_2 v^{k-1} + (1 - \beta_2)(\nabla f_{S_k}(x^k) \odot \nabla f_{S_k}(x^k))$

5:      Compute $\mathbf{D}_k = \mathrm{diag}(\varepsilon\mathbf{I} + \sqrt{v^k/(1 - \beta_2^k)})$

6:      Compute

$$\gamma_k = \frac{\frac{c}{(1+\lambda c)}\left[1 - \frac{c\lambda}{2f_{S_k}(x^k)}\nabla f_{S_k}(x^k)^\top x^k\right]_+}{1 + \frac{c}{2f_{S_k}(x^k)(1+\lambda c)}\|\nabla f_{S_k}(x^k)\|_{\mathbf{D}_k^{-1}}^2}$$

7:      Update $x^{k+1} = \frac{1}{1+\lambda c}x^k - (1 - \beta_1)\gamma_k \mathbf{D}_k^{-1}\nabla f_{S_k}(x^k) + \beta_1(x^k - x^{k-1})$

8: **end for**

    $[\cdot]_+$ denotes $\max\{0, \cdot\}$.

---

Therefore, we have

$$p^k = -\left(\frac{c}{1+\lambda c}\mathbf{\Sigma}_k^{-1} - \frac{\frac{2c^2}{(1+\lambda c)^2}\mathbf{\Sigma}_k^{-1}\nabla r(x^k)\nabla r(x^k)^\top\mathbf{\Sigma}_k^{-1}}{1 + \frac{2c}{1+\lambda c}\nabla r(x^k)\mathbf{\Sigma}_k^{-1}\nabla r(x^k)}\right)(2r(x^k)\nabla r(x^k) + \lambda\mathbf{\Sigma}_k x^k)$$

$$= -\frac{2cr(x^k)}{1+\lambda c}\left(1 - \frac{\frac{2c}{1+\lambda c}\nabla r(x^k)^\top\mathbf{\Sigma}_k^{-1}\nabla r(x^k)}{1 + \frac{2c}{1+\lambda c}\nabla r(x^k)\mathbf{\Sigma}_k^{-1}\nabla r(x^k)}\right)\mathbf{\Sigma}_k\nabla r(x^k)$$

$$\quad - \frac{\lambda c}{1+\lambda c}x^k + \frac{\frac{2c^2\lambda}{1+\lambda c}\mathbf{\Sigma}_k^{-1}\nabla r(x^k)\nabla r(x^k)^\top x^k}{1 + \frac{2c}{1+\lambda c}\nabla r(x^k)\mathbf{\Sigma}_k^{-1}\nabla r(x^k)}$$

$$= -\frac{2cr(x^k)}{1+\lambda c}\frac{1}{1 + \frac{2c}{1+\lambda c}\nabla r(x^k)\mathbf{\Sigma}_k^{-1}\nabla r(x^k)}\mathbf{\Sigma}_k^{-1}\nabla r(x^k)$$

$$\quad - \frac{\lambda c}{1+\lambda c}x^k + \frac{\frac{2c^2\lambda}{1+\lambda c}\mathbf{\Sigma}_k^{-1}\nabla r(x^k)\nabla r(x^k)^\top x^k}{1 + \frac{2c}{1+\lambda c}\nabla r(x^k)\mathbf{\Sigma}_k^{-1}\nabla r(x^k)}.$$

Using the connection $\nabla r(x^k) = \frac{1}{2\sqrt{f(x^k)}}\nabla f(x^k)$ and $r(x^k) = \sqrt{f(x^k)}$ we get

$$p^k = -\frac{2c\sqrt{f(x^k)}}{1+\lambda c}\frac{1}{1 + \frac{2c}{4f(x^k)(1+\lambda c)}\nabla f(x^k)^\top\mathbf{\Sigma}_k^{-1}\nabla f(x^k)}\mathbf{\Sigma}_k^{-1}\frac{1}{2\sqrt{f(x^k)}}\nabla f(x^k)$$

$$\quad - \frac{c\lambda}{1+\lambda c}x^k + \frac{\frac{2c^2\lambda}{4f(x^k)(1+\lambda c)}\mathbf{\Sigma}_k^{-1}\nabla f(x^k)\nabla f(x^k)^\top x^k}{1 + \frac{2c}{4(1+\lambda c)f(x^k)}\nabla f(x^k)^\top\mathbf{\Sigma}_k^{-1}\nabla f(x^k)}$$

$$= -\frac{c/(1+\lambda c)}{1 + \frac{c}{2f(x^k)(1+\lambda c)}\|\nabla f(x^k)\|_{\mathbf{\Sigma}_k^{-1}}^2}\mathbf{\Sigma}_k\nabla f(x^k) - \frac{c\lambda}{1+\lambda c}x^k$$

$$\quad + \frac{c\lambda}{1+\lambda c}\frac{\frac{c}{2f(x^k)}\nabla f(x^k)^\top x^k}{1 + \frac{c}{2f(x^k)(1+\lambda c)}\|\nabla f(x^k)\|_{\mathbf{\Sigma}_k^{-1}}^2}\mathbf{\Sigma}_k^{-1}\nabla f(x^k).$$

To summarize, the update of NGN-Dv1W is the following

$$x^{k+1} = x^k + p^k$$

$$= \frac{1}{1+\lambda c}x^k + \frac{c\lambda}{1+\lambda c}\frac{\frac{c}{2f(x^k)}\nabla f(x^k)^\top x^k}{1+\frac{c}{2f(x^k)(1+\lambda c)}\|\nabla f(x^k)\|^2_{\boldsymbol{\Sigma}_k^{-1}}}\boldsymbol{\Sigma}_k^{-1}\nabla f(x^k)$$

$$-\frac{c/(1+\lambda c)}{1+\frac{c}{2f(x^k)(1+\lambda c)}\|\nabla f(x^k)\|^2_{\boldsymbol{\Sigma}_k^{-1}}}\boldsymbol{\Sigma}_k^{-1}\nabla f(x^k)$$

$$= \frac{1}{1+\lambda c}x^k - \frac{\frac{c}{1+\lambda c}\left(1-\frac{c\lambda}{2f(x^k)}\nabla f(x^k)^\top x^k\right)}{1+\frac{c}{2f(x^k)(1+\lambda c)}\|\nabla f(x^k)\|^2_{\boldsymbol{\Sigma}_k^{-1}}}\boldsymbol{\Sigma}_k^{-1}\nabla f(x^k). \tag{36}$$

To prevent the step-size next to $\boldsymbol{\Sigma}_k^{-1}\nabla f(x^k)$ from being negative, the final update has the form

$$x^{k+1} = \frac{1}{1+\lambda c}x^k - \frac{\frac{c}{1+\lambda c}\left[1-\frac{c\lambda}{2f(x^k)}\nabla f(x^k)^\top x^k\right]_+}{1+\frac{c}{2f(x^k)(1+\lambda c)}\|\nabla f(x^k)\|^2_{\boldsymbol{\Sigma}_k^{-1}}}\boldsymbol{\Sigma}_k^{-1}\nabla f(x^k), \tag{37}$$

where $[\cdot]_+ := \max\{\cdot, 0\}$. Now we can add momentum on top and obtain the following update of NGN-MDv1W

$$x^{k+1} = \frac{1}{1+\lambda c}x^k - \frac{\frac{c}{1+\lambda c}\left[1-\frac{c\lambda}{2f(x^k)}\nabla f(x^k)^\top x^k\right]_+}{1+\frac{c}{2f(x^k)(1+\lambda c)}\|\nabla f(x^k)\|^2_{\boldsymbol{\Sigma}_k^{-1}}}\boldsymbol{\Sigma}_k^{-1}\nabla f(x^k) + \beta(x^k - x^{k-1}). \tag{38}$$

This combination of NGN-MDv1 and weight decay is summarized in Algorithm 4. We highlight that now the weight decay is incorporated inside the adaptive step-size as well as regularizing the coefficient next to $x^k$.

## F.2 EMPIRICAL VALIDATION OF THE PROPOSED COMBINATIONS

Having two possible ways of adding weight decay to NGN-MDv1, we test them on pretraining a 70M transformer on language modeling. The validation perplexity at the end of training is reported in Figure 6. We note that when weight decay is turned off, both NGN-MDv1W and Dec-NGN-MDv1 reduce to NGN-MDv1.

First, we observe that when weight decay is properly tuned, all algorithms improve over the baseline case with no weight decay, which is consistent with the observation of Xiao (2024) and Andriushchenko et al. (2023) on AdamW. We also note that Dec-NGN-MDv1 and NGN-MDv1W require a smaller weight decay value compared to the other algorithms. Finally, the stability and performance of NGNMDv1 are preserved by both variations, allowing training with larger learning rates, and significantly improving over AdamW and Momo-Adam.

We do not observe a substantial difference between the two proposed modifications of NGN-MDv1 for this task. We remark however that these two versions serve substantially different purposes, and pretraining language models might not be the most representative task to evaluate the effect of adding regularization.

## G ADDITIONAL EXPERIMENTS AND TRAINING DETAILS

### G.1 TRAINING DETAILS

The detailed experiment setup with hyper-parameters and training details is presented in Table 2. We provide links to the exact model architectures used in our experiments (the links are clickable) as well as links to the tables and figures for each workload. We demonstrate the results averaged across 3 different random seeds for small and middle-range size experiments. We use standard values of momentum parameters $(\beta_1, \beta_2) = (0.9, 0.999)$ if the opposite is not specified. The step-size hyper-parameter is tuned across powers of 10 (for some workloads we add additional values of the step-size hyper-parameter shown in the step-size resilience plots). We use PyTorch (Paszke et al., 2017) implementation of Adam. The implementation of MomSPS, Momo, Momo-Adam are

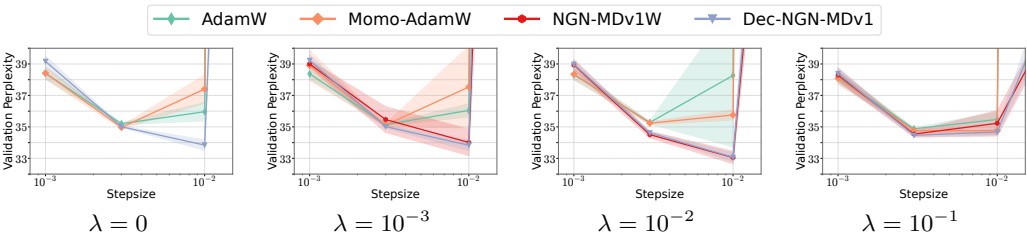

Figure 6: Adding weight decay when pretraining a 70M Transformer++. When properly tuned, a value of weight decay $> 0$ enhances the performance of all algorithms. NGN-MDv1 retains his characteristic stability, and achieves smaller perplexity in all scenarios.

provided in the corresponding papers. Finally, when employing SGD-M, we set dampening equal to 0.9.

For vision transformers experiments, we follow the setup of Schaipp et al. (2024), and use Pytorch Image Models codebase (Wightman, 2019). We train a `vit_tiny_patch16_224` for 200 epochs on Imagenet1k, using a cosine learning rate schedule with a linear warmup of 5 epochs. Differently than Schaipp et al. (2024), we train in `bfloat16`, instead of `float16`, and do not employ weight decay regularization.

For pre-training Transformers on Causal Language Modeling, we build upon the nanoGPT (Karpathy, 2022) implementation, augmenting it with Rotational Positional Embedding (Su et al., 2023), RMSNorm (Zhang & Sennrich, 2019), and SwiGLU (Shazeer, 2020). We call this enhanced version Transformer++. Models are trained with a batch size of 256, context length of 2048 tokens, vocabolary size of 50280 and make use of GPT-Neox tokenizer (Black et al., 2022). We adopt an enhanced training recipe, made popular by large language models such as LLaMa (Touvron et al., 2023). These modifications include: training in `bfloat16`; employing a linear learning rate warmup for 10% of the training steps, followed by cosine annealing to $10^{-5}$; omitting biases from linear layers; using $(\beta_1, \beta_2) = (0.9, 0.95)$ for all algorithms; clipping gradient norms above 1; no weight tying between embedding and last linear layer. All models are trained on SlimPajama-627B (Soboleva et al., 2023), a cleaned and deduplicated version of RedPajama We report validation perplexity on a separate subset of Slim-Pajama consisting of 10M tokens. The total compute is estimated following Kaplan et al. (2020), where the estimated number of floating-point operations (FLOPs) is 6 × Number of Parameters × Number of Tokens.

Experiments of small and middle size are performed on 1xRTX 4090. We perform ImageNet32 experiments on 2xA100-40GB, and ImageNet1k experiments on 4xA100-SXM4-40GB. For pretraining Transformers on Language Modeling, we employ 8xH100-HBM3-80GB GPUs. With multiple devices in use, we employ Distributed Data Parallel to parallelize the training process.

### G.2 COMPARISON ALGORITHMS THAT SUPPORT MOMENTUM

In the main paper, we provided the test performance only. Now we additionally illustrate the performance of algorithms w.r.t. training loss convergence. Figure 7 demonstrates that NGN-M is the most robust algorithm for the choice of the step-size hyper-parameter from this perspective as well. In Figure 7, we additionally demonstrate the performance of the algorithms on (VGG16 (Simonyan & Zisserman, 2014), CIFAR10) and (MLP, MNIST) workloads where NGN-M matches the performance of the state-of-the-art algorithms in this setting and archives higher resilience to the step-size hyper-parameter choice. The best performance results are reported in Table 3 and showcase that NGN-M always matches the performance of other optimizers or improves it.

### G.3 COMPARISON OF ALGORITHMS THAT SUPPORT MOMENTUM AND DIAGONAL STEP-SIZE

Next, we illustrate the performance of the algorithms that support both momentum and diagonal step-size. According to the results in Figures 8 and 9, NGN-MDv1 achieves the best resilience to the step-size hyper-parameter choice among all considered algorithms. Again, NGN-MDv1 is the most

Table 2: Summary of experiment setup with all the details on hyper-parameters used in each case.

| Model | Dataset | Performance Results | Stability Results | Effective Stepsize Results | Epochs / Iterations | Batch Size | Comments |
|---|---|---|---|---|---|---|---|
| Resnet20 | CIFAR10 | Tab. 3, 4, 5 | Fig. 2, 7, 8, 11 | Fig. 14, 15, 22 | 50 | 128 | |
| Resnet110 | CIFAR100 | Tab. 3, 4 | Fig. 2, 7, 8, 12 | | 100 | 128 | |
| VGG16 | CIFAR10 | Tab. 3, 4 | Fig. 7, 8 | | 50 | 128 | |
| MLP | MNIST | Tab. 3, 4 | Fig. 7, 9 | | 10 | 128 | 2 hidden layers of size 100 |
| ViT | CIFAR10 | Tab. 3, 4 | Fig. 2, 7, 8, 13 | Fig. 5, 14, 15, 23 | 200 | 512 | |
| LSTM | PTB | Tab. 4, 5 | Fig. 9 | | 150 | 20 | 3 layers |
| LSTM | Wikitext-2 | Tab. 4, 5 | Fig. 24 | | 150 | 20 | 3 layers |
| Transformer | Rotten Tomatoes | Tab. 4, 5 | Tab. 24 | | 2000 | 16 | # heads 8 # layers 24 |
| Transformer | Tiny Shakespeare | Tab. 4, 5 | Fig. 9, 24 | | 2000 | 16 | # heads 8 # layers 24 |
| Resnet18 | ImageNet32 | Tab. 3, 4, | Fig. 10 | | 45 | 128 | constant learning rate schedule; no weight decay |
| Resnet18 | ImageNet1k | Tab. 3, 4 | Fig. 2, 10 | | 90 | 256 | learning rate decay every 30 epochs by 0.1 no weight decay |
| ViT-Tiny | ImageNet1k | Tab. 4 | Fig. 3 | | 200 | 512 | cosine learning rate schedule with linear warm-up for 5 epochs no weight decay, `bfloat16` |
| 70M Transformer++ | SlimPajama-627B | Tab. 4 | Fig. 4, 6 | | 2400 | 256 | dim=512, # heads 8 # layers 6, context length 2048 $(\beta_1, \beta_2) = (0.9, 0.95)$, `bfloat16` clipping norm 1, linear warm-up for 10% of iterations |
| 160M Transformer++ | SlimPajama-627B | Tab. 4 | Fig. 4 | | 4800 | 256 | dim=768, # heads 12 # layers 12, context length 2048 $(\beta_1, \beta_2) = (0.9, 0.95)$, `bfloat16` clipping norm 1, linear warm-up for 10% of iterations |
| 420M Transformer++ | SlimPajama-627B | Tab. 4 | Fig. 4, 21 | | 13500 | 256 | dim=1024, # heads 16 # layers 24, context length 2048 $(\beta_1, \beta_2) = (0.9, 0.95)$, `bfloat16` clipping norm 1, linear warm-up for 10% of iterations |

Table 3: The best validation score (with one standard deviation across 3 runs; accuracy for computer vision tasks; perplexity for NLP tasks) for the best learning rate choice for each method that supports momentum.

| Model | Dataset | NGN | SGDM | NGN-M | MomSPS | Momo | ALR-SMAG |
|---|---|---|---|---|---|---|---|
| Resnet20 | CIFAR10 | $88.30_{\pm0.20}$ | $85.42_{\pm0.70}$ | $88.76_{\pm0.05}$ | $87.20_{\pm0.38}$ | $88.86_{\pm0.14}$ | $88.88_{\pm0.19}$ |
| Resnet110 | CIFAR100 | $64.76_{\pm0.26}$ | $57.16_{\pm2.06}$ | $64.98_{\pm0.29}$ | $63.37_{\pm0.71}$ | $64.81_{\pm0.33}$ | $64.73_{\pm1.81}$ |
| VGG16 | CIFAR10 | $90.21_{\pm0.10}$ | $89.67_{\pm0.43}$ | $90.42_{\pm0.06}$ | $87.26_{\pm0.21}$ | $90.43_{\pm0.17}$ | $90.49_{\pm0.35}$ |
| MLP | MNIST | $98.04_{\pm0.07}$ | $97.63_{\pm0.10}$ | $97.97_{\pm0.08}$ | $97.73_{\pm0.09}$ | $97.97_{\pm0.04}$ | $97.64_{\pm0.06}$ |
| ViT | CIFAR10 | $83.34_{\pm0.24}$ | $83.74_{\pm0.11}$ | $84.95_{\pm0.29}$ | $83.77_{\pm0.27}$ | $85.47_{\pm0.27}$ | $85.54_{\pm0.39}$ |
| Resnet18 | ImageNet32 | 48.63 | 48.56 | 48.29 | N/A | 48.68 | N/A |
| Resnet18 | ImageNet1k | 67.00 | 66.73 | 67.12 | N/A | 67.09 | N/A |
| Transformer | Tiny Shakespeare | $9.27_{\pm0.19}$ | $8.73_{\pm0.13}$ | $7.67_{\pm0.12}$ | N/A | $8.80_{\pm0.19}$ | N/A |
| Transformer | Rotten Tomatoes | $9.01_{\pm0.22}$ | $8.75_{\pm0.04}$ | $7.12_{\pm0.03}$ | N/A | $8.65_{\pm0.03}$ | N/A |
| LSTM | Wikitext-2 | $75.33_{\pm0.15}$ | $82.07_{\pm0.16}$ | $75.51_{\pm0.22}$ | N/A | $76.09_{\pm0.40}$ | N/A |

stable algorithm to the choice of step-size hyper-parameter w.r.t. training loss convergence. Its best performance is competitive to that of other algorithms but the step-size hyper-parameter range that gives such performance is wider.

Moreover, we support our claims about stability on additional workloads such as (VGG16, CIFAR10) (in Figure 7), (MLP, MNIST), (LSTM (Hochreiter & Schmidhuber, 1997), PTB (Mikolov et al., 2010)), and (Transformer (Karpathy, 2022), Tiny Shakespeare (Karpathy, 2015)) workloads. We observe that NGN-MDv1 attains higher robustness to the choice of the step-size hyper-parameter. Finally, the performance results on (LSTM, Wikitext-2 (Merity et al., 2016)) and (Transformer, Rotten Tomatoes (Pang & Lee, 2005)) are reported in Table 4. The results demonstrate competitive performance of NGN-MDv1 against other benchmarks across all considered workloads.

Table 4: The best validation score (with one standard deviation; accuracy for computer vision tasks; perplexity for NLP tasks) for the best learning rate choice for each method that supports diagonal step-sizes and momentum.

| Model | Dataset | Adam | Momo-Adam | NGN-MDv1 | NGN-MDv2 | Lion | Adabelief | Adabound |
|---|---|---|---|---|---|---|---|---|
| Resnet20 | CIFAR10 | $86.96_{\pm0.70}$ | $89.41_{\pm0.36}$ | $89.53_{\pm0.11}$ | $87.80_{\pm0.16}$ | $88.09_{\pm0.27}$ | $87.47_{\pm0.48}$ | $85.00_{\pm0.56}$ |
| Resnet110 | CIFAR100 | $64.12_{\pm0.94}$ | $67.10_{\pm0.53}$ | $66.10_{\pm0.45}$ | $64.33_{\pm0.40}$ | $61.85_{\pm0.77}$ | $65.32_{\pm0.43}$ | $61.28_{\pm0.39}$ |
| VGG16 | CIFAR10 | $90.26_{\pm0.23}$ | $90.95_{\pm0.28}$ | $90.64_{\pm0.18}$ | $90.07_{\pm0.37}$ | N/A | N/A | N/A |
| MLP | MNIST | $97.44_{\pm0.19}$ | $97.96_{\pm0.10}$ | $98.10_{\pm0.06}$ | $97.67_{\pm0.17}$ | N/A | N/A | N/A |
| ViT | CIFAR10 | $85.96_{\pm0.23}$ | $85.74_{\pm0.12}$ | $85.65_{\pm0.10}$ | $86.56_{\pm0.11}$ | $86.89_{\pm0.19}$ | $85.05_{\pm0.47}$ | $80.32_{\pm0.47}$ |
| Transformer | Rotten Tomatoes | $6.80_{\pm0.07}$ | $6.81_{\pm0.05}$ | $6.90_{\pm0.05}$ | $6.83_{\pm0.05}$ | N/A | N/A | N/A |
| Transformer | Tiny Shakespeare | $6.80_{\pm0.06}$ | $6.80_{\pm0.05}$ | $6.89_{\pm0.06}$ | $6.82_{\pm0.05}$ | N/A | N/A | N/A |
| LSTM | PTB | $70.95_{\pm0.08}$ | $71.09_{\pm0.05}$ | $70.84_{\pm0.20}$ | $71.37_{\pm0.17}$ | N/A | N/A | N/A |
| LSTM | Wikitext-2 | $81.49_{\pm1.49}$ | $82.23_{\pm0.64}$ | $75.24_{\pm0.21}$ | $81.99_{\pm0.78}$ | N/A | N/A | N/A |
| Resnet18 | ImageNet32 | 48.11 | 48.09 | 48.06 | 47.55 | N/A | N/A | N/A |
| Resnet18 | ImageNet1k | 67.17 | 67.06 | 67.15 | 67.32 | N/A | N/A | N/A |
| ViT-Tiny | ImageNet1k | $71.05_{\pm0.16}$ | $71.22_{\pm0.36}$ | $71.345_{\pm0.22}$ | N/A | N/A | N/A | N/A |
| Transformer++ 70M | SlimPajama-627B | $35.20_{\pm0.06}$ | $34.96_{\pm0.11}$ | $33.84_{\pm0.33}$ | N/A | N/A | N/A | N/A |
| Transformer++ 160M | SlimPajama-627B | $24.26_{\pm0.10}$ | $24.29_{\pm0.10}$ | $23.42_{\pm0.10}$ | N/A | N/A | N/A | N/A |
| Transformer++ 420M | SlimPajama-627B | 17.00 | 17.07 | 16.60 | N/A | N/A | N/A | N/A |

### G.4 ADDITIONAL IMAGENET EXPERIMENTS

Now we turn to the experiments involving training Resnet18 on ImageNet1k and ImageNet32. In Figure 10 we provide the train loss curves and results on (Resnet18, ImageNet32) workload that demonstrate that NGN-M and NDN-MDv1 attain better resilience to the step-size hyper-parameter choice than competitors not only from the train loss point of view as well. The best performance of algorithms is provided in Table 3 and 4. According to them, both NGN-M and NGN-M achieve competitive performance against considered benchmarks.

### G.5 ADDITIONAL COMPARISON AGAINST LION, ADABELIEF, ADABOUND

This section compares algorithms from Section 5.1 and Section 5.2. Moreover, we include the comparison against Lion (Chen et al., 2024), Adabound (Luo et al., 2019), and Adabelief (Zhuang et al., 2020). The results are presented in Table 4.

We observe that NGN-MDv1 and NGN-MDv2 both achieve competitive performance across various Deep Learning workloads. In Figures 11 to 13, we observe that Lion, Adabound and Adabelief algorithms do not match always the performance of NGN-MDv1 and Adam: Adabelief has worse performance on (Resnet20, CIFAR10) workload; Adabound has worse performance on (Resnet20, CIFAR10), (Resnet110, CIFAR100), and (ViT, CIFAR10) workloads; Lion has worse performance on (Resnet110, CIFAR100) workload. Moreover, their resilience to the step-size hyper-parameter choice is lower than that of NGN-MDv1. To summarize, NGN-M and NGN-MDv1 are the most robust algorithms to the choice of step-size hyper-parameter.

### G.6 COMPARISON OF ALGORITHMS WITH DIAGONAL STEP-SIZE

Now we compare algorithms with diagonal step-size such as NGN-D, Adagrad Duchi et al. (2011), and RMSprop Kingma & Ba (2015). Since NGN-D requires to find constants $\{c_j\}_{j=1}^d$ where $d$ is the size of the model. Finding sufficiently good constants $c_j$ might be a challenging task since $d$ is a large number. Therefore, we use RMSprop preconditioner $\mathbf{D}_k$ to set them as $c_j = c/(\mathbf{D}_k)_{(j)}$. We leave the exploration of how to set constants $c_j$ properly for future research.

For each method, we tune its learning rate hyper-parameter over the powers of 10: $\{10^{-4}, \ldots, 10^2\}$ and present the best performance averaged across 3 random seeds in Table 5. We observe that NGN-D performs similarly to RMSprop. NGN-D has slightly worse performance on (LSTM, PTB)

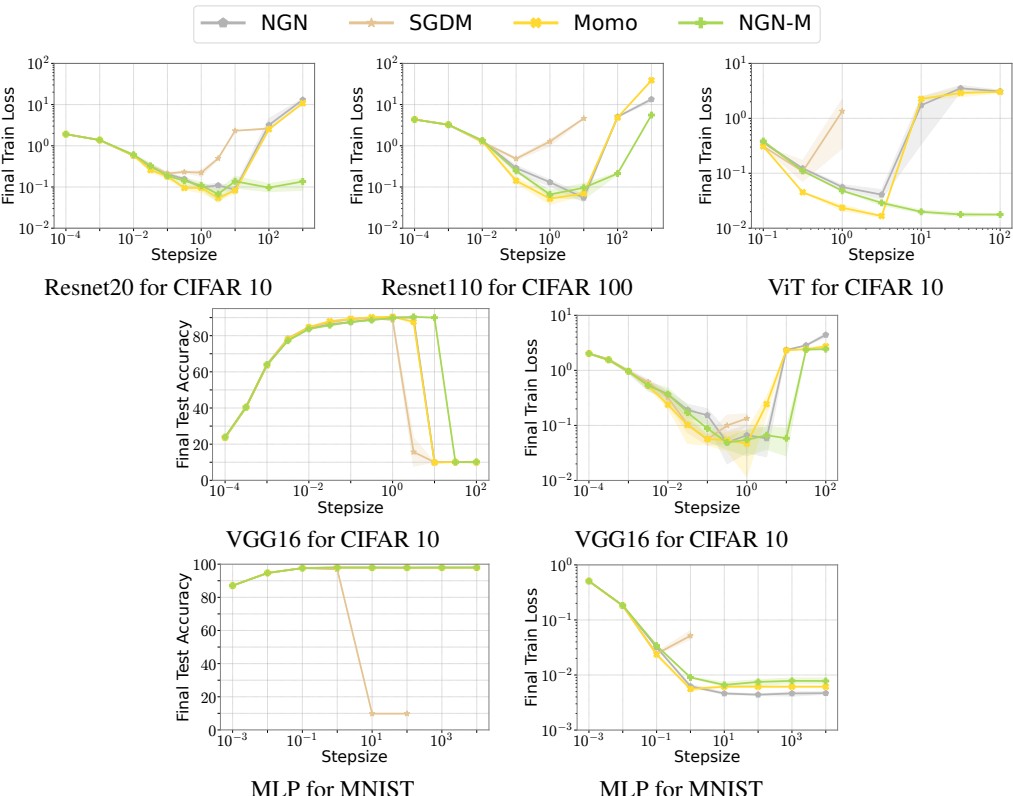

Figure 7: Stability performance of algorithms supporting momentum varying step-size hyper-arameter ($c$ for NGN and NGN-M, $\alpha_0$ for Momo, and step-size for SGDM). We observe that NGN-M achieves the training loss close to the best possible for a wider range of the step-size hyper-parameter.

Table 5: The best validation score (with one standard deviation; accuracy for image classification; perplexity for language modeling) for the best learning rate choice for each method that supports diagonal step-sizes.

| Model | Dataset | Adagrad | RMSprop | NGN-D |
|---|---|---|---|---|
| Resnet20 | CIFAR10 | $85.90_{\pm 0.30}$ | $86.71_{\pm 0.64}$ | $86.98_{\pm 0.15}$ |
| Transformer | Rotten Tomatoes | $7.77_{\pm 0.02}$ | $6.87_{\pm 0.05}$ | $6.92_{\pm 0.03}$ |
| Transformer | Tiny Sheaksper | $7.77_{\pm 0.05}$ | $7.00_{\pm 0.13}$ | $6.90_{\pm 0.05}$ |
| LSTM | PTB | $99.24_{\pm 2.13}$ | $69.00_{\pm 0.17}$ | $71.54_{\pm 0.11}$ |
| LSTM | Wikitext-2 | $113.19_{\pm 4.36}$ | $79.48_{\pm 0.45}$ | $75.44_{\pm 0.12}$ |

dataset but significantly better on (LSTM, Wikitext-2) workload. Besides, Adagrad always has the worst performance. Moreover, these algorithms do not have high resilience to the choice of hyper-parameter. Therefore, we omit their comparison from this perspective.

### G.7 EFFECTIVE STEP-SIZE OF NGN-M, Momo, NGN-MDv1, AND Momo-Adam

Next, we compare the effective step-size applied throughout the training with NGN-M, Momo, NGN-MDv1, and Momo-Adam in Figures 14 and 15. First, both NGN-M and Momo perform a warm-up in the beginning: the effective step-size increases at the beginning of the training. Then we observe the main difference between the two algorithms above: effective step-size of Momo for sufficiently large step-size hyper-parameter is not adaptive within some part of the training, it always hits the upper bound. Consequently, during that part of the training Momo reduces to SGDM. In contrast, the effective step-size of NGN-M is always adaptive: it gradually decreases after a short warm-up. This

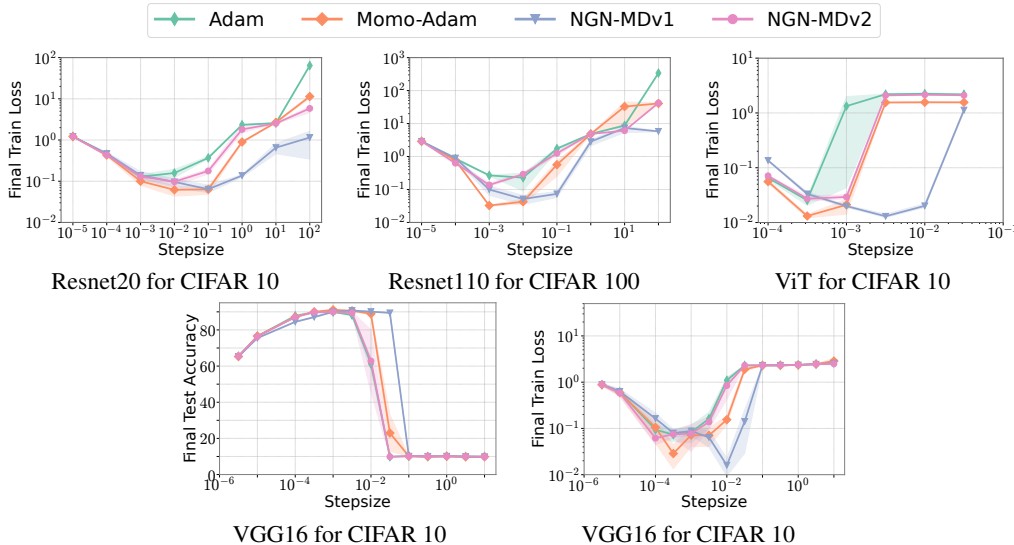

Figure 8: Stability performance of algorithms supporting momentum and diagonal step-size varying step-size hyper-parameter ($c$ for NGN-MDv1 and NGN-MDv2, $\alpha_0$ for Momo-Adam, and step-size for Adam). We observe that NGN-MDv1 achieves the training loss close to the best possible for a wider range of the step-size hyper-parameter.

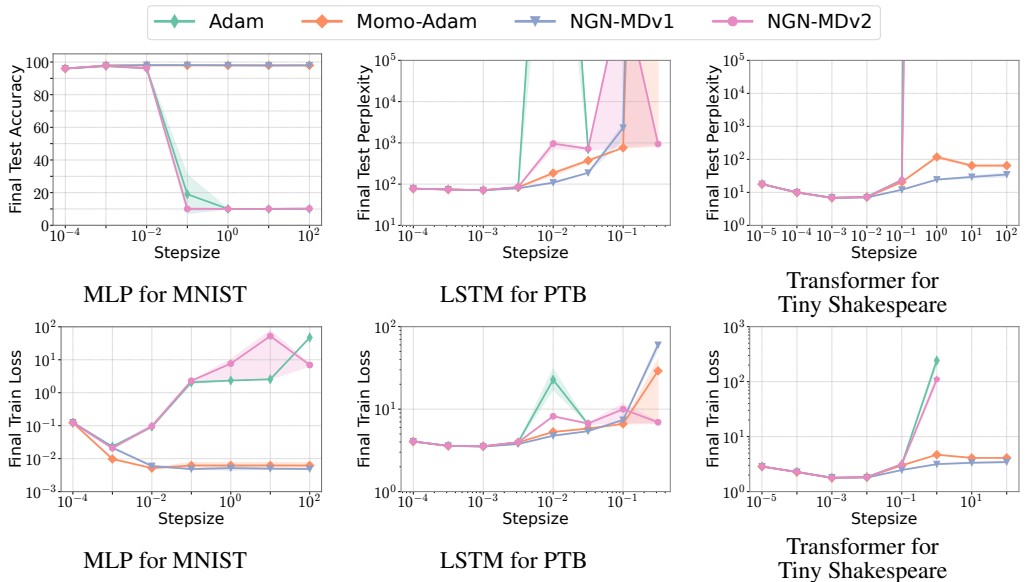

Figure 9: Stability performance of algorithms supporting momentum and diagonal step-size varying step-size hyper-parameter ($c$ for NGN-MDv1 and NGN-MDv2, $\alpha_0$ for Momo-Adam, and step-size for Adam). We observe that NGN-MDv1 achieves the training loss close to the best possible for a wider range of the step-size hyper-parameter.

trend is similar to the state-of-the-art learning rate schedulers used in practice. Similar observations can be made in comparison of NGN-MDv1 and Momo-Adam.

### G.8   SPECTRUM EVOLUTION DURING THE TRAINING WITH NGN-M AND SGDM

We include the results that demonstrate the spectrum evolution of the training and test losses in the training of Resnet20. From Figure 19 and Figure 20, we see that

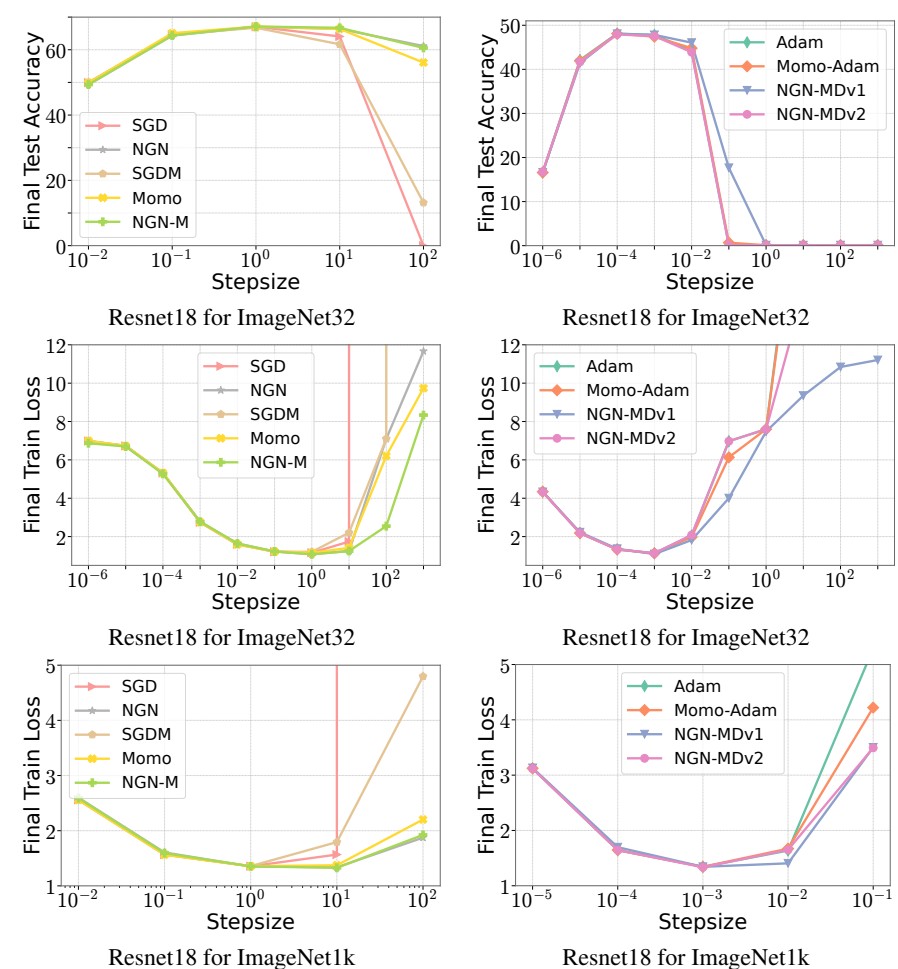

Figure 10: Stability performance of algorithms supporting momentum (**first row**), and momentum with diagonal step-size (**second row**) varying step-size hyper-parameter ($c$ for NGN, NGN-M, NGN-MDv1, and NGN-MDv2, $\alpha_0$ for Momo and Momo-Adam, and step-size for SGD, SGDM, and Adam).

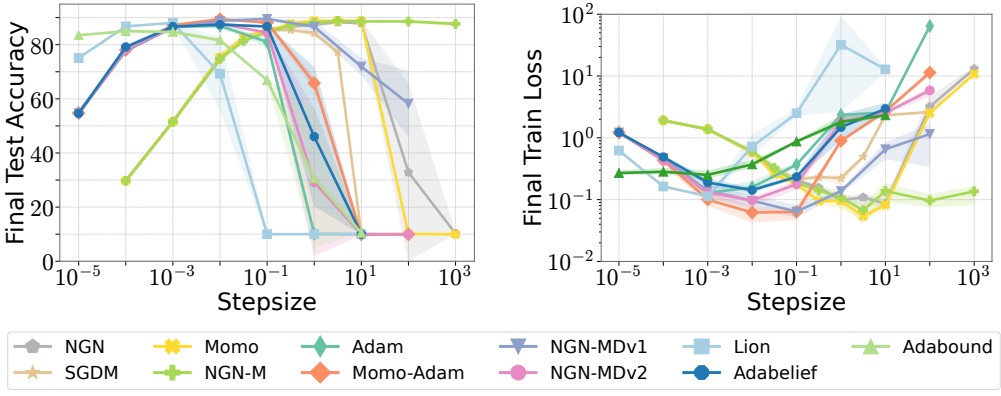

Figure 11: Stability performance of various optimizers for Resnet20 on CIFAR10.

1. Both NGN-M and SGDM increase the sharpness throughout the training for small step-size hyper-parameter values ($c \in \{10^{-4}, 10^{-3}\}$). The sharpness throughout the training has similar values for NGN-M and SGDM.

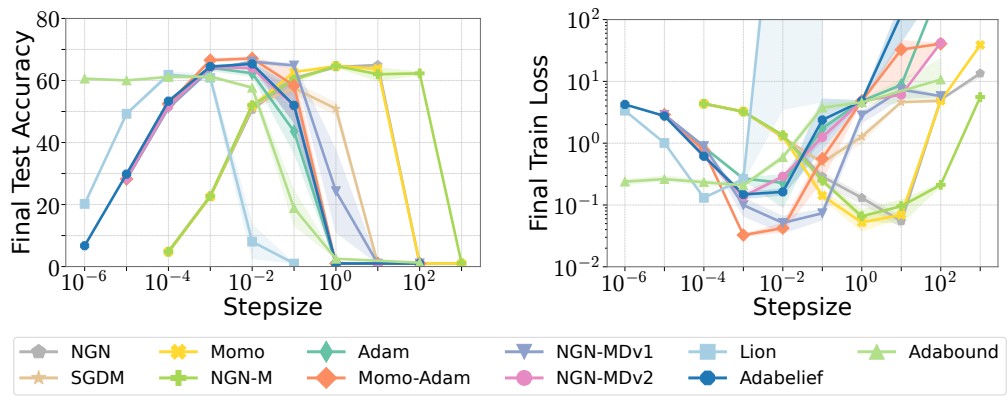

Figure 12: Stability performance of various optimizers for Resnet110 on CIFAR100.

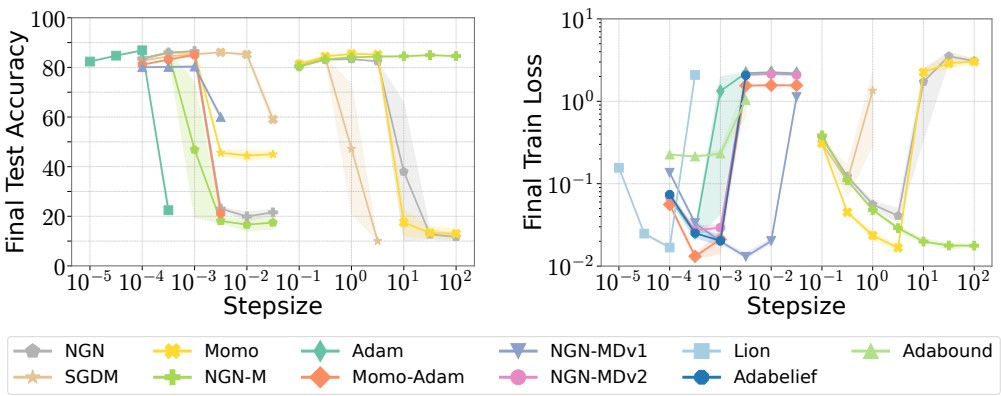

Figure 13: Stability performance of various optimizers for ViT on CIFAR10.

2. Both NGN-M and SGDM increase the sharpness in the beginning of the training, and decrease to the end of the training for middle-range values of $c \in \{10^{-2}, 10^{-1}\}$. The sharpness throughout the training has similar values for NGN-M and SGDM.

3. Both NGN-M and SGDM decrease the sharpness for $c = 10^0$. However, the sharpness throughout the training with NGN-M is higher than that with SGDM.

4. NGN-M converges with $c = 10^1$ step-size hyper-parameter while SGDM fails. Moreover, the sharpness throughout the training with NGN-M with $c = 10^1$ is smaller than that with smaller values of $c \in \{10^{-4}, \ldots, 10^0\}$. This result suggests that for large enough values of $c$ NGN-M tends to converge to flatter minima with increasing the step-size hyper-parameter. Being in flat minima allows the use of large effective step-sizes as the function value does not increase much there.

These observations are also supported from the loss landscape perspective along top two eigenvectors; see Figures 16 and 17.

The observed phenomenon is strongly related to training at the Edge of Stability (EoS), as explored in Cohen et al. (2021) and other studies. However, we emphasize that Cohen et al. (2021) focuses on non-adaptive methods, both with and without momentum. The only work we are aware of that examines EoS behavior in adaptive methods is Cohen et al. (2022). According to Cohen et al. (2022), Adam operates at an adaptive EoS (determined by the eigenvalues of the preconditioned Hessian), even as standard sharpness continues to increase throughout training. Our findings indicate that NGN-M operates at the Edge of Stability (EoS), despite employing adaptive step sizes.

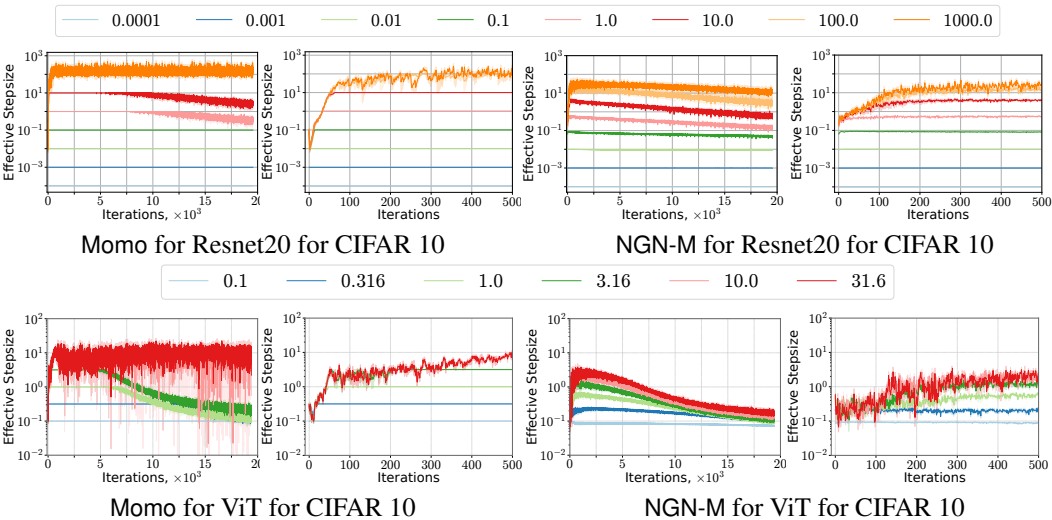

Figure 14: The step-size of Momo and NGN-M during the training. We demonstrate the step-sizes $\tau_k$ for Momo and $\gamma_k$ for NGN-M varying step-size parameters $\alpha_0$ for Momo and $c$ for NGN-M.

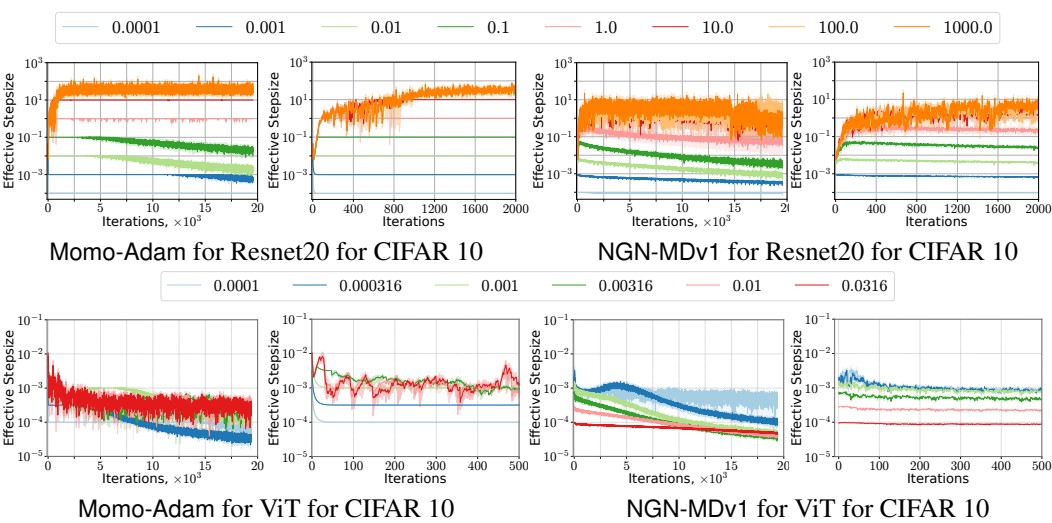

Figure 15: The step-size of Momo-Adam and NGN-MDv1 during the training. We demonstrate the step-sizes $\tau_k$ for Momo-Adam and $\gamma_k$ for NGN-MDv1 varying step-size parameters $\alpha_0$ for Momo and $c$ for NGN-MDv1.

## G.9 COMPARISON OF ADAPTIVE STEP-SIZES OF Adam, Momo-Adam, AND NGN-MDv1

Next, we conduct experiments to compare the adaptive step-size of Adam, Momo-Adam, and NGN-MDv1. Note that ResNet20 model consists of 3 base blocks, and each block has 3 convolution layers. In Figure 22 we plot the average adaptive step-size of the layers $j \in$ {layer1.0.conv1, layer2.0.conv1, layer3.0.conv1} of ResNet20 that corresponds to the first convolution layer within each base block. Similarly, in Figure 23 we plot the average adaptive step-size of the layers $j \in$ {layer0.0.fn.to_qkv, layer3.0.fn.to_qkv, layer5.0.fn.to_qkv} that corresponds to the attention layers of the first, fourth, and sixth base blocks.

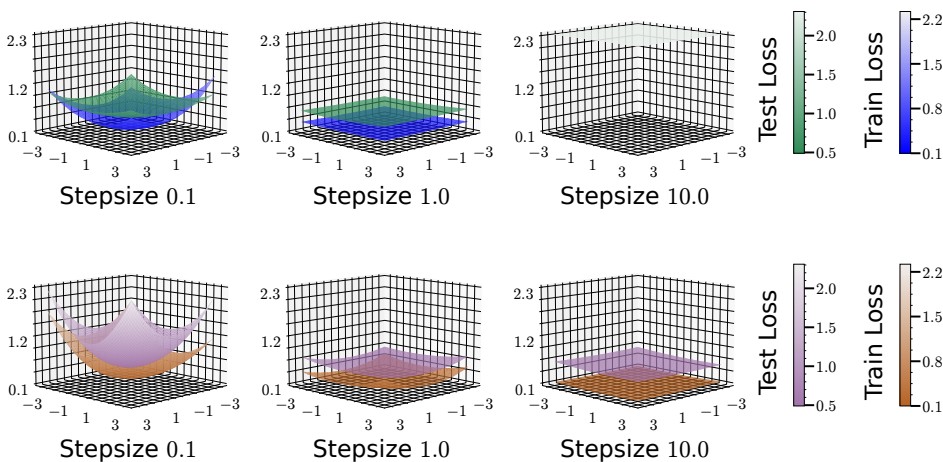

Figure 16: Loss landscape of train (lower surface) and test (upper surface) losses along two largest unit eigenvectors around the last iterate of SGDM (**first row**) and NGN-M (**second row**) for Resnet20 on CIFAR10 for a fixed random seed.

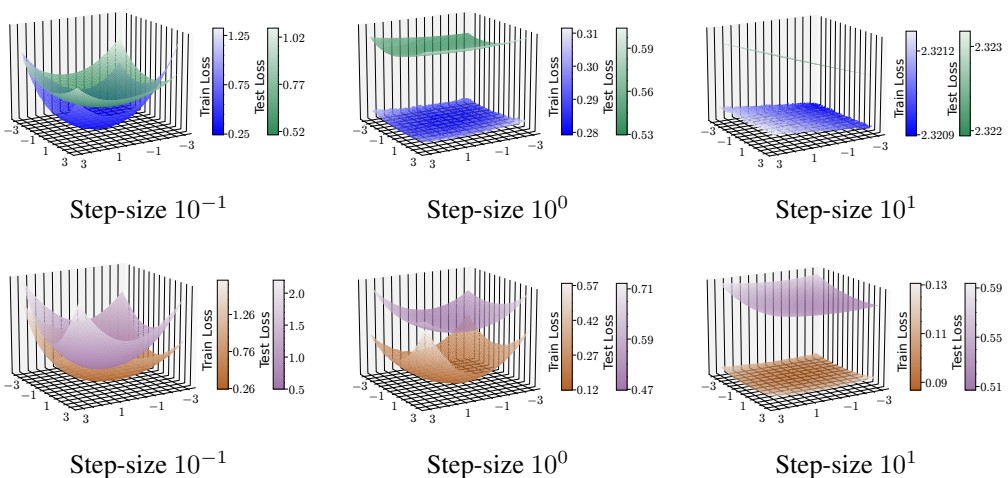

Figure 17: Zoomed loss landscape of train and test losses along top eigenvectors around the last iterate of SGDM (**first row**) and NGN-M (**second row**) for Resnet20 on CIFAR10. The change of test and train losses is given in the color bars. We observe that increasing the step-size hyper-parameter for NGN-M leads to convergence to flatter minima while for SGDM it leads to divergence.

Since the adaptivity of Adam is only in the second-order momentum applied as a normalization, in our experiment we compare the following quantities

$$\frac{\gamma}{(\mathbf{D}_k)_{(j)}} \text{ for Adam}, \quad \frac{\tau_k}{(\mathbf{D}_k)_{(j)}} \text{ for Momo-Adam}, \quad \frac{\gamma_k}{(\mathbf{D}_k)_{(j)}} \text{ for NGN-MDv1}, \quad (39)$$

where $\gamma$ is the step-size hyper-parameter of Adam.

Let us first describe the results for ResNet20 in Figure 22. We observe that NGN-MDv1 tends to set smaller effective step-size compared to two other algorithms. This is especially visible for the large step-size hyper-parameter values where the adaptive step-size of NGN-MDv1 is by several orders in magnitude smaller than that of Adam and Momo-Adam. In contrast, the coordinate-wise adaptive step-size of Momo-Adam is mostly follow that of Adam. Considering that the stability performance of NGN-MDv1 is much higher for this task, this happens mainly due to the fact that the adaptation mechanism of NGN-MDv1 step-size is more conservative than that of Momo-Adam.

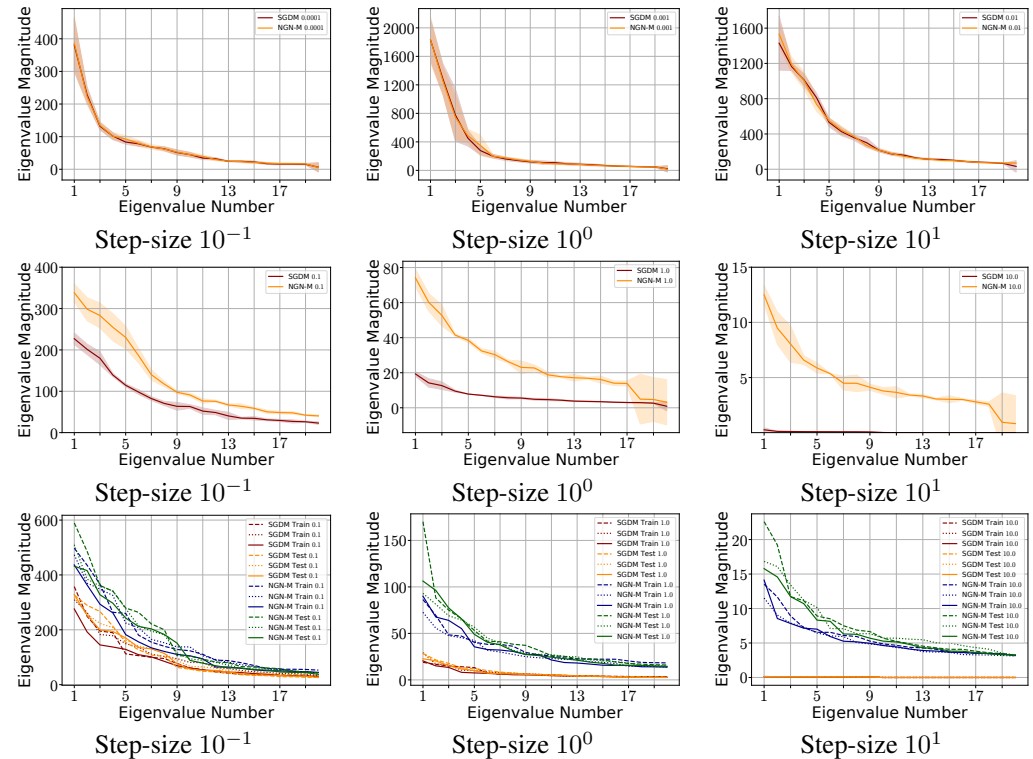

Figure 18: **First and Second rows**: 20 largest train eigenvalues averaged across 3 runs after 50 epochs of training with SGDM and NGN-M of Resnet20 on CIFAR10. **Third row**: 20 largest train and test eigenvalues for 3 different random seeds after 50 epochs of training with SGDM and NGN-M of Resnet20 on CIFAR10.

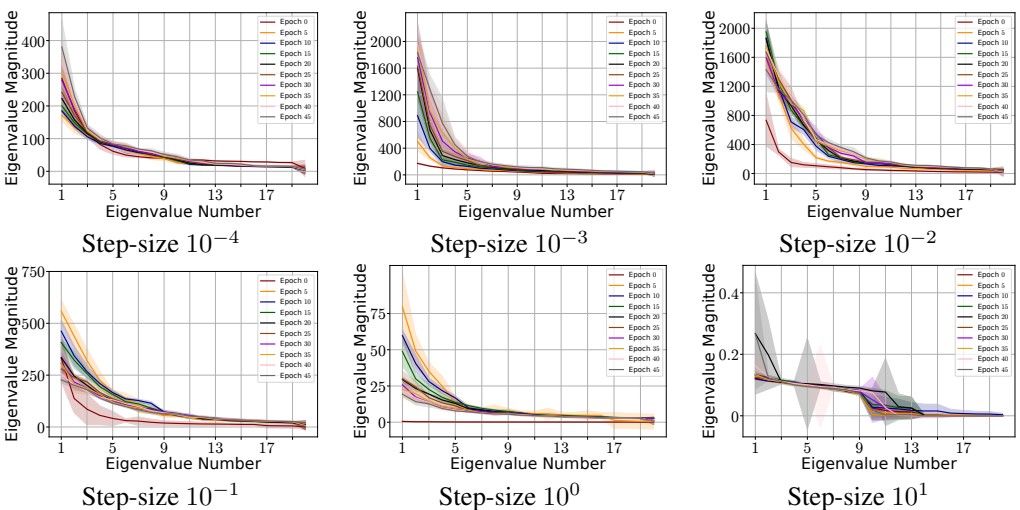

Figure 19: The evolution of 20 largest train eigenvalues averaged across 3 runs during the training with SGDM of Resnet20 on CIFAR10.

Now we switch to the results on ViT model in Figure 23. Here both Momo-Adam and NGN-MDv1 tend to utilize smaller effective coordinate-wise step-size, by several orders in magnitude smaller than that of Adam. However, the adaptation mechanism of NGN-MDv1 is still more conservative than that of Momo-Adam, especially for large step-size hyper-parameters. We also highlight that in this experiment the best performance of NGN-MDv1 is achieved with $c = 10^{-3}$. When we vary the

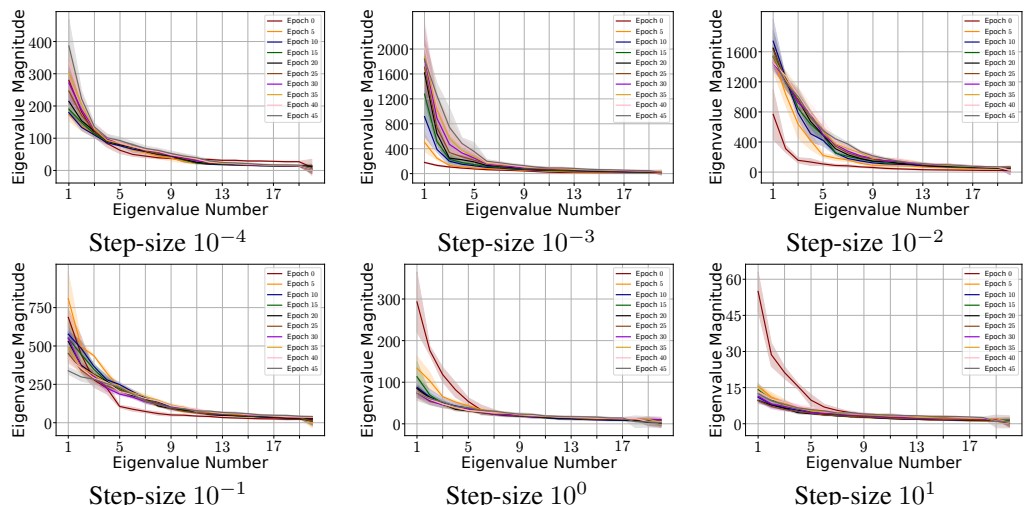

Figure 20: The evolution of 20 largest train eigenvalues averaged across 3 runs during the training with NGN-M of Resnet20 on CIFAR10.

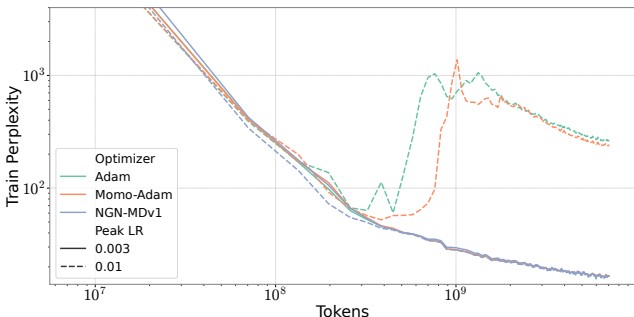

Figure 21: Training dynamics of a 420M Transformer+++ trained on SlimPajama. NGN-MDv1 achieves lowest perplexity, and, unlike Adam and Momo-Adam, it remains stable at a larger learning rate of 0.01.

step-size hyper-parameter $c$, the effective coordinate-wise step-size does not change dramatically, especially for layers.0.0.fin.to_qkv layer.

## G.10 Extended Comparison of Momentum-based Algorithms on NLP Tasks

We switch to comparison of NGN-M, Momo, NGN, and SGDM on NLP tasks. In particular, we consider the training of Transformer (based on NanoGPT) on the Tiny Shakespeare and Rotten Tomatoes datasets and LSTM on the Wikitext-2 dataset from Appendix G.3. We report the results in Figure 24 while the best performance is shown in Table 3. First, note that all algorithms do not match the best performance of those that incorporate diagonal step-size and momentum (see Table 4). Such results are expected since the training of NLP models has significantly different coordinate-wise conditioning. Nonetheless, NGN-M algorithm achieves better resilience to the step-size hyper-parameter choice, especially in the training of Transformer models. Therefore, NGN-M across various model architectures and task domains.

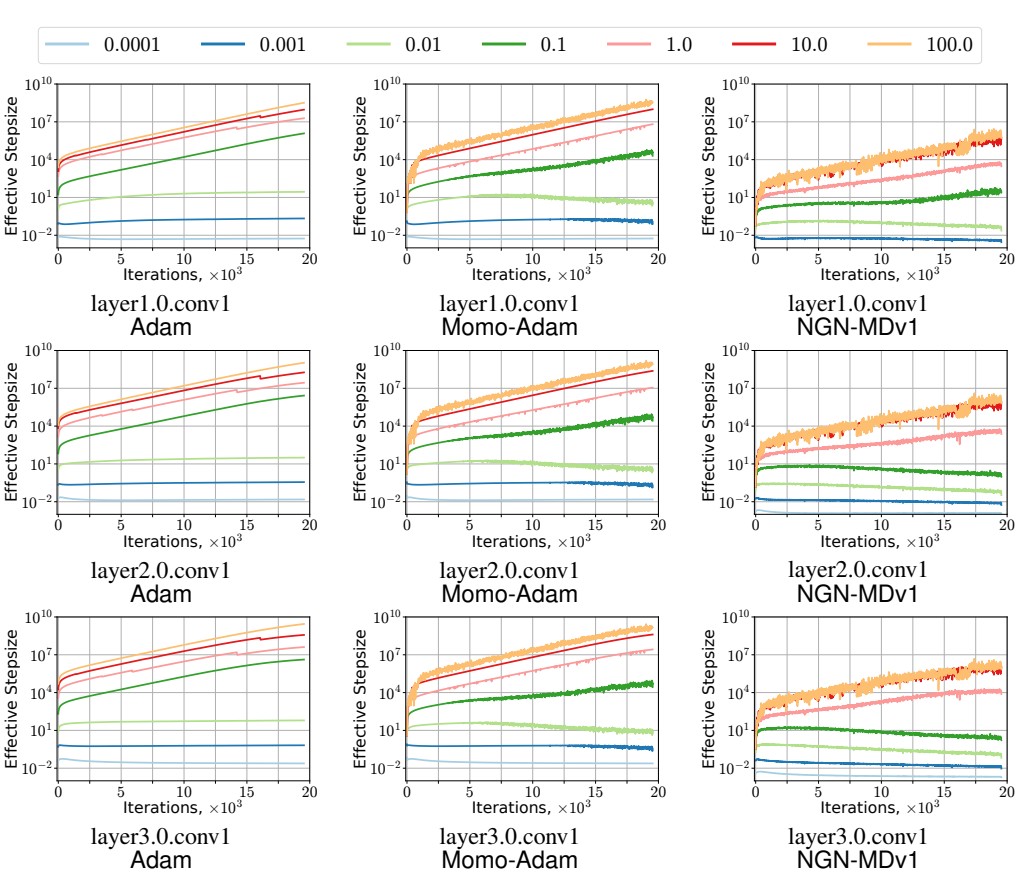

Figure 22: The adaptive stepsize of Adam (**first column**), Momo-Adam (**second column**), and NGN-MDv1 (**third column**) algorithms in training ResNet20 model on CIFAR10 dataset. We plot the average stepsize $\frac{\gamma}{(\mathbf{D}_k)_{(j)}}$ (for Adam), $\frac{\tau_k}{(\mathbf{D}_k)_{(j)}}$ (for Momo-Adam), and $\frac{\gamma_k}{(\mathbf{D}_k)_{(j)}}$ (for NGN-MDv1) for the first convolution layer within each of 3 base blocks of ResNet20 architecture varying the stepsize hyper-parameter of the algorithms ($c$ for NGN-M and NGN, $\alpha_0$ for Momo, and learning rate parameter for Adam).

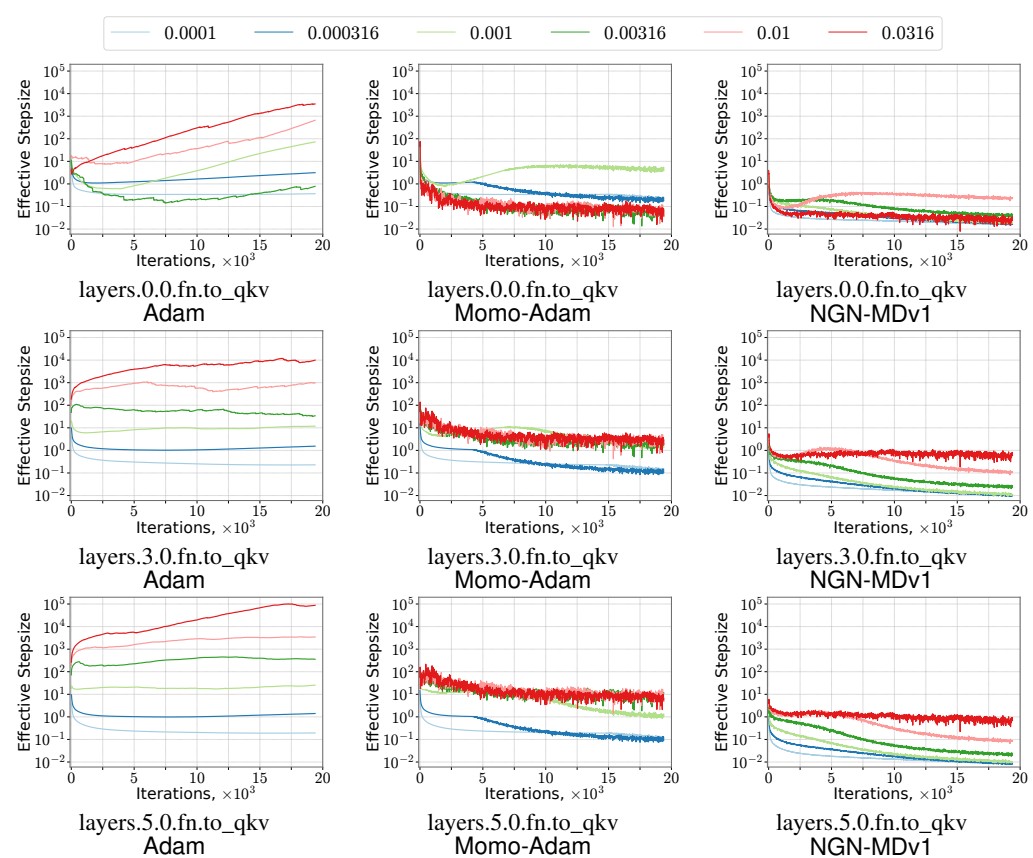

Figure 23: The adaptive stepsize of Adam (**first column**), Momo-Adam (**second column**), and NGN-MDv1 (**third column**) algorithms in training ViT model on CIFAR10 dataset. We plot the average stepsize $\frac{\gamma}{(\mathbf{D}_k)_{(j)}}$ (for Adam), $\frac{\tau_k}{(\mathbf{D}_k)_{(j)}}$ (for Momo-Adam), and $\frac{\gamma_k}{(\mathbf{D}_k)_{(j)}}$ (for NGN-MDv1) for the attention layer within each of the first, fourth, and sixth base blocks of ViT architecture varying the step-size hyper-parameter of the algorithms ($c$ for NGN-M and NGN, $\alpha_0$ for Momo, and learning rate parameter for Adam).

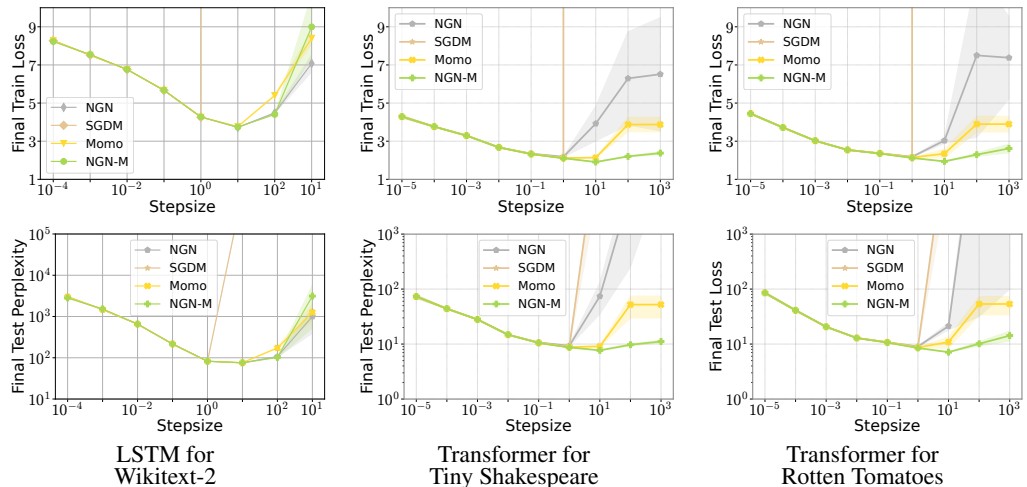

Figure 24: Stability performance of algorithms supporting momentum and diagonal step-size varying step-size hyper-parameter ($c$ for NGN-M and NGN, $\alpha_0$ for Momo, and step-size for SGDM). We observe that NGN-M achieves the training loss close to the best possible for a wider range of the step-size hyper-parameter.

