# OpenReview forum: "Enhancing Optimizer Stability: Momentum Adaptation of NGN Step-size"
_ICLR.cc/2025/Conference — Submitted to ICLR 2025_

### Official Review · Reviewer_KwKA · 2024-11-02

**Soundness:** 3
**Presentation:** 3
**Contribution:** 3
**Rating:** 8
**Confidence:** 3

**Summary:**

The paper’s main contribution is the development of an adaptive algorithm for the NGN step size, termed NGN-M, incorporating momentum to improve robustness in hyper-parameter selection. NGN-M addresses the sensitivity issues in hyper-parameter choices (specifically step-size). The authors provide a theoretical analysis, ensuring that the new adaptive momentum-based NGN (NGN-M) achieves a convergence rate of $O(1/\sqrt(K))$ under convex settings, and mention that this rate even without common assumptions like interpolation or bounded gradients. The authors not only propose the NGN-M but also introduce two variants—NGN-MD targeted at enhancing robustness in step size through a diagonal adaptation approach.

**Strengths:**

The robustness of the proposed NGN-M and NGN-MD, in dealing with a wide range of step sizes is a major strength. This is a practical advantage, as it reduces the need for extensive hyperparameter tuning. They compared their method clearly with other related methods in the literature in Table 1. They provided theorems that clearly state assumptions and convergence rates for their method. They justify that their assumption for theorem 1 is commonly made in the literature. They support their theoretical findings with experimental results(e.g., image and language models) highlighting NGN-M's performance's applicability and potential as a general-purpose optimizer.

**Weaknesses:**

There is a bounded variance assumption in Theorem 2 which may limit practical applicability in cases where these conditions do not hold (e.g. RL setting).
However, I have not seen significant weaknesses in this paper. I should mention that I have not read the proofs in the appendix.

**Questions:**

Have you done any further exploration of the non-convex settings and NGN’s potential there?

---

> ### Author Response · Authors · 2024-11-21
> **Rebuttals**
>
> **W1:** We would like to highlight that our convergence guarantees are provided under the assumption that the interpolation error $\sigma^2_{\rm int} := \mathbb{E}[f^* - f_i^*]$ is bounded. This assumption is satisfied if each of the used loss functions is bounded below. This is typically the case for most of the standard losses (MSE, logistic loss, cross-entropy).
>
> **Q1:** Thank you for this question. Please note that the convergence of NGN-D in the general non-convex regime and under PL is provided in the appendix. The convergence guarantees match those of SGD. However, we highlight that the convergence of NGN-D is a more challenging task since it requires handling the adaptive stepsize. This challenge is overcome in the similar way as it was done for NGN-M: the effective step-size is split into two parts: a deterministic and a stochastic part. In our analysis, this decomposition of the step-size $\gamma_k$ enables us to regulate the balance between the
> descent term, which drives improvement in the objective, and the error term, which reflects possible
> inaccuracies. More precisely, the descent term is weighted by a constant $c$ while the error term proportional to $\sigma^2_{\rm int}$ is weighted by $c^2$, which suggests that $c$ can be chosen to trade off the two terms to lead to the exact convergence similarly to the standard analysis of SGD.
>
> Nonetheless, we would like to draw the reviewer's attention to the fact NGN-M reduces to SGDM when the step-size hyper-parameter
> $c\to 0$ (see discussion in Section 2.2.3 in [1] for more details). Therefore, informally speaking the convergence should be expected for a small enough step-size hyper-parameter but it requires a careful analysis to demonstrate the convergence formally.
>
> [1] Orvieto, Antonio and Xiao, Lin, An adaptive stochastic gradient method with non-negative gauss-newton stepsizes, arXiv preprint arXiv:2407.04358, 2024.

---

> > ### Comment · Reviewer_KwKA · 2024-11-27
> >
> > I appreciate your thorough response and intend to keep my positive score for your paper.

---

### Official Review · Reviewer_Qgdv · 2024-11-02

**Soundness:** 2
**Presentation:** 3
**Contribution:** 2
**Rating:** 5
**Confidence:** 2

**Summary:**

The paper introduces NGN-M, an optimizer that combines NGN step-size with momentum, achieving robust, state-of-the-art performance with less sensitivity to step-size tuning. Theoretical analysis shows convergence comparable to SGDM in convex settings, and experiments on CIFAR and ImageNet validate its effectiveness and stability across varying hyperparameters. Two additional variants, NGN-D and NGN-MD, further extend its adaptability, making NGN-M a promising approach for stable, efficient optimization in deep learning.

**Strengths:**

1. **Enhanced Stability**: NGN-M addresses a key limitation in adaptive optimizers—sensitivity to learning rates—by combining momentum and NGN step-size. This stability improvement is backed by both theoretical and empirical results.

2. **Theoretical Contributions**: The authors provide rigorous theoretical analysis showing that NGN-M’s convergence rate matches that of SGDM in convex settings. This is a notable achievement since it removes the bounded gradients constraint common in other optimizers.

3. **Extensive Empirical Validation**: Results on diverse tasks, including ResNet and Vision Transformers on CIFAR datasets and ImageNet, demonstrate NGN-M’s robustness and competitive performance. Additionally, the optimizer performs well even at larger step sizes, which is uncommon among traditional optimizers.

**Weaknesses:**

1. **Limited Non-Convex Analysis**: While the authors focus on convex settings, optimizers like Adam are widely used in non-convex scenarios, such as deep networks. Addressing NGN-M’s performance under non-convex conditions would strengthen its applicability.

2. **Complexity of the NGN-MD Variant**: NGN-MD, which uses both momentum and diagonal preconditioning, can be computationally intensive due to additional matrix operations. In practice, this may pose challenges for large-scale applications, especially in memory-constrained environments.

3. **Lack of Comparison on NLP Tasks**: Although NGN-M shows promise in vision tasks, the paper could benefit from an expanded evaluation on NLP tasks, where optimizers like Adam and AdamW are dominant.

**Questions:**

As above

---

> ### Author Response · Authors · 2024-11-21
> **Rebuttals**
>
> **W1:** We agree that deriving convergence guarantees in the general non-convex setting is of significant interest. We emphasize that the NGN-D algorithm has been analyzed in the non-convex regime, both in the general case and under the Polyak-Łojasiewicz condition. However, analyzing NGN-M in the non-convex case is considerably more challenging than in the convex setting, so we have decided to leave this analysis for future work. Nonetheless, we would like to draw the reviewer's attention to the fact NGN-M reduces to SGDM when the step-size hyper-parameter $c \to 0$ (see discussion in Section 2.2.3 in [1] for more details). Therefore, informally speaking the convergence should be expected for a small enough step-size hyper-parameter but it requires a careful analysis to demonstrate the convergence formally.
>
> [1] Orvieto, Antonio and Xiao, Lin, An adaptive stochastic gradient method with non-negative gauss-newton stepsizes, arXiv preprint arXiv:2407.04358, 2024.
>
> **W2:** Implementing NGN-MDv1 in practice might be slightly more computationally expensive. However, we highlight that computing NGN-MDv1 step does not involve matrix-vector operations since the preconditioner is a diagonal matrix, and the matrix notation is used only for the convenience of presentation. The additional computation cost that we have in NGN-MDv1 is the computation of $\\|\nabla f_{S_k}(x^k)\\|^2_{\mathbf{D}\_k^{-1}}$. This can be done by one pass over the gradient and summing the terms $\frac{1}{(\mathbf{D}\_k)_{j}}(\nabla_j f\_{S_k}(x^k))^2$ for $j\in[d].$ This operation does not require additional matrix multiplication and can be computed while updating $\mathbf{D}\_k$. The rest of the NGN-MDv1 implementation does not add any significant costly operations in comparison with Adam. Besides, the step of NGN-MDv2 does not change much as well since it does not even require the computation of the gradient norm in the weighted norm. We add this discussion about this in the revised version in Section E.2.
>
> **W3:** Thank you for raising this point. We agree with the reviewer that NLP tasks are of significant interest. We in fact already provided such experiments in the original draft and as we will discuss below, we have added additional experimental results to address this comment.
>
> Before we elaborate further on these experimental results, it is important to highlight that training NLP models presents distinct challenges compared to vision models. Specifically, NLP tasks often require coordinate-wise adaptive step-sizes, making the NGN-M variant (as well as Momo and SGDM) less suitable for these tasks. Developing coordinate-adaptive methods is therefore a critical research direction for improving NLP model training in the future.
>
> Following the previous argument, we demonstrate the performance of only NGN-MD against Momo-Adam and Adam in pretraining large scale language models since NGN-M would not match their best performance. We stress that the pretraining is the most resource intensive, general purpose and challenging NLP task in deep learning. Regarding other experimental results provided in the paper, we refer the reviewer to Figure 9 (additional experiments on LSTM and Transformer models), which might have been overlooked as it is located in the appendix. This figure provides a comparison of NGN-MDv1 and NGN-MDv2 against Momo-Adam and Adam, and demonstrates the good performance of the NGN variants against other methods. In addition, we have extended our experiments in the updated version of the pdf to include comparisons of NGN-M, Momo, SGDM, and NGN in training LSTM and Transformer models. The stability plots are presented in Figure 24 while the detailed discussion is given in Section G.10. Moreover, the best performance of the algorithms is reported in Table 3. From these results, we can draw two conclusions:
>
> - **Robustness to Step-Size Selection:** NGN-M achieves better resilience to the step-size hyper-parameter choice for these settings than other algorithms which is in line with the results in the main paper.
> - **Performance of Momentum-Based Algorithms:** As expected the performance of momentum-based algorithms does not match that of algorithms that use both momentum and diagonal step-size. This is due to the significantly different conditioning across coordinates in NLP tasks, emphasizing the need for further research into advanced adaptive coordinate-wise step-size methods.
>
>
> These results underscore the strengths of the NGN variants while highlighting opportunities for future work in the area of NLP models.

---

> ### Author Response · Authors · 2024-11-23
> **Reminder**
>
> Dear reviewer,
>
> We would like to remind you that the discussion period ends soon. Therefore, we would like to know if there are concerns left unaddressed or needed to be clarified. We would be happy to discuss them further.

---

### Official Review · Reviewer_D8is · 2024-11-03

**Soundness:** 3
**Presentation:** 2
**Contribution:** 3
**Rating:** 6
**Confidence:** 3

**Summary:**

This paper builds upon a variant of the Polyak step-size called NGN (Non-negative Gauss-Newton) and offers two main variants.

The first algorithm combines NGN with momentum, resulting in the NGN-M algorithm. The authors theoretically prove that NGN-M achieves a convergence rate of $O(1/\sqrt{K})$ in the convex setting, without requiring assumptions of bounded gradients or interpolation.

The second variant is inspired by prevalent coordinate-wise adaptive optimization methods like Adam for training neural networks and introduces an NGN for coordinate-wise step-size configuration in the NGN-D algorithm. The authors show that NGN-D converges with a rate of $O(1/\sqrt{K})$ for convex and smooth functions with bounded noise variance. They use this coordinate-wise variant, NGN-D, with momentum to create the NGN-MD algorithm. When used with RMSprop preconditioning, they refer to it as NGN-MDv1, otherwise NGN-MDv2.

Empirically, NGN-M and NGN-MD achieve comparable performance with baselines and demonstrate enhanced robustness to step-size selection in experiments on CIFAR10, CIFAR100, ResNet18, and ViT.

**Strengths:**

Developing new adaptive optimization methods that match Adam’s performance while providing additional stability and convergence guarantees for certain problem classes represents a significant contribution to the current deep learning literature.

**Weaknesses:**

These are relatively minor issues. Overall, I believe the paper would benefit from an additional pass for writing clarity.

- The difference between NGN-MDv1 and NGN-MDv2 is not explained in text, and it is confusing. I would generally prefer if you treated NGN-MDv1 and NGN-MDv2 similarly to how you approached ver1 and ver2 of NGN-M: by experimenting to determine which performs better and then using that version consistently throughout the paper.

- The organization of the NGN-D section in Section 3.3, along with Algorithm 2, could be improved. For example, in Equation 3, you already have $\gamma_k$ multiplied by $\Sigma$, and then you choose $\Sigma^{-1}_j = \gamma^j$, which results in a squared $\gamma_k$ as the step-size. Based on Algorithm 2, I don’t think this is what you intended.

- Figure 5 could also be improved: please add method names more clearly. Could you also compare the effective step-size of Adam? I would be more interested in that comparison than in one with MoMo, as MoMo is not shown in the ViT experiments. In that experiment, you compare MoMo-Adam, Adam, and NGN-MDv1, so it would make more sense to focus on those methods.

**Questions:**

- Is it obvious that all NGN-based methods perform better than SPS (stochastic Polyak step-size)? If not, did you consider adding it to the baselines for the NGN-M algorithm?

- Your analysis of NGN-M in Theorem 1 notably avoids the need for the interpolation condition, which is often a key assumption for proving convergence in stochastic Polyak step-size methods. Given that interpolation occurs in overparameterized neural networks, I found it a reasonable assumption. Could you elaborate on the specific mechanisms that enable the proof of convergence rates without relying on the interpolation condition?

---

> ### Author Response · Authors · 2024-11-21
> **Rebuttals**
>
> **W1:** Thank you for this insightful comment. We plan to include a more detailed comparison between the two versions of NGN-MD in the revised manuscript. Below is the discussion we incorporated in Section E.1.
>
> Both algorithms use the RMSprop preconditioner $\mathbf{D}\_k$ that performs an exponential moving average of the coordinate-wise squared gradient norm. Moreover, both algorithms use momentum to perform an averaging of the preconditioned updates $\boldsymbol{\Sigma}_k^{-1}\nabla f\_{S_k}(x^k).$
> Nonetheless, there are two key differences between the proposed algorithms. First, $\mathbf{D}\_k$ is used as a preconditioner directly in NGN-MDv1 while in NGN-MDv2 it is used to rescale the $c$ constant for each coordinate inside coordinate-wise NGN step-size. Second, NGN-MDv1 uses one global NGN step-size weighted by the $\mathbf{D}_k$ norm while in NGN-MDv2 we use coordinate-wise NGN step-size replacing the full gradient by the corresponding partial derivative. According to the empirical results, both versions with a tuned step-size hyperparameter are competitive with other baselines but NGN-MDv1 demonstrates much better stability performance.
>
> The main difference in comparison with Adam is the order in which the preconditioning and momentum are applied. In both NGN-MDv1 and NGN-MDv2 we average the preconditioned updates $\boldsymbol{\Sigma}\_k^{-1}\nabla f_{S_k}(x^k),$ i.e. we first apply preconditioning and momentum later. In contrast, in Adam and Momo-Adam the stochastic gradients are averaged to construct a new momentum term, and then the momentum is preconditioned. In other words, the momentum is applied first and then it is followed by preconditioning. We believe this change might be one of the reasons behind the step-size hyper-parameter resilience of NGN-MD.
>
> In practice, we found out that the tuned performance of NGN-MDv1 is slightly better than that of NGN-MDv2. Moreover, NGN-MDv1 demonstrates higher resilience to the choice of the step-size hyper-parameter than NGN-MDv2.
>
> **W2:** We thank the reviewer for this comment. We rewrote this section in the revised version. Nonetheless, we would like to clarify the raised concerns. The derivations in (2) are used to provide an intuition of how one can add a diagonal step-size into NGN by choosing the weight matrix $\boldsymbol{\Sigma}\_k$. These derivations are used exactly in the NGN-MDv1 algorithm with $\boldsymbol{\Sigma}\_k = \mathbf{D}\_k$ where $\mathbf{D}\_k$ is a RMSprop preconditioner. In this case, we have only one global NGN step-size in front of $\mathbf{D}\_k$. For NGN-D the derivations follow a more straightforward intuition. We can update each parameter $j$ using the coordinate-wise NGN step-size where the gradient norm $\\|\nabla f\_{S_k}(x^k)\\|$ is replaced by the corresponding partial derivative $|\nabla\_j f\_{S_k}(x^k)|.$ Namely, each coordinate is updated using $\gamma_k^{(j)} = \frac{c}{1+\frac{c}{2f_{S_k}(x^k)}(\nabla_j f_{S_k}(x^k))^2}$. To derive the NGN-MDv2 algorithm from NGN-D, we observe that each parameter requires a coordinate-wise NGN step-size hyperparameter $c$. To achieve this, we use the RMSprop preconditioner to set the coordinate-wise NGN step-size as $c/(\mathbf{D}\_k)\_{(j)}$. Incorporating a momentum on top leads to the NGN-MDv2 algorithm.

---

> ### Author Response · Authors · 2024-11-21
> **Rebuttals (Part 2)**
>
> **W3:** We will add the names of the algorithms in the caption of each plot in Figure 5 to make the figure more readable. In Figure 5, the two left plots correspond to the comparison of Momo and NGN-M while the two right plots correspond to the comparison of Momo-Adam and NGN-MDv1. We would like to highlight that the performance of the Momo algorithm for the (ViT, CIFAR10) setting is presented in the top row of Fig. 2.
>
> Following the request from the reviewer, we provide the comparison of the adaptive step-size of Adam, Momo-Adam, and NGN-MDv1 in Section G.9 for two different settings: (ResNet20, CIFAR10) and (ViT, CIFAR10); see a more detailed discussion there as well. Note that Adam's adaptive step-size relies solely on normalization using the second-order momentum while both Momo-Adam and NGN-MDv1 have an additional step-size ($\tau_k$ and $\gamma_k$ respectively) factor in addition to the normalization. Therefore, we decided to compare
> $$\frac{\gamma}{(\mathbf{D}\_k)\_{(j)}} \text{ for Adam},\quad \frac{\tau_k}{(\mathbf{D}\_k)\_{(j)}} \text{ for Momo-Adam}, \quad \frac{\gamma_k}{(\mathbf{D}\_k)\_{(j)}} \text{ for NGN-MDv1},$$
> where $j\in[d]$ corresponds to some specific parameter of the model. We visualize the average quantities defined above for some specific layers of models (i.e., we average the effective step-size of algorithms for all $j$ within some layer). In particular, we present the results for the first convolution layer of each base block of ResNet20, and for the attention layer of $1,3,6$ base blocks of ViT model. We observe that in both experiments the effective coordinate-wise step-size of NGN-MDv1 is smaller than for the other two optimizers. In other words, the adaptive step-size of NGN-MDv1 is more conservative and does not allow the effective step-size to increase too much even when the step-size hyper-parameter is set to be large. This demonstrates that the NGN-MDv1 is less sensitive to the choice of the step-size hyper-parameter while allowing it to reach comparable or superior performance to other optimizers.
>
> **Q1:** In fact, the main difference between the NGN and SPS${}\_{\max}$ step-sizes is that the NGN step-size is a harmonic mean between the constant step-size of SGD and SPS. On the opposite, SPS${}\_{\max}$ is the minimum between the constant step-size of SGD and SPS. The harmonic averaging can be seen as a soft version of the minimum. Although the difference does not appear to be highly significant at first, it allows NGN to be both theoretically and practically superior. We refer to [1], sec. 2.2.3 for a more detailed discussion on the differences between NGN and SPS step-sizes. We exclude a comparison against MomSPS as its best performance is almost always worse than those of NGN-M and Momo (see Tab 3).  Moreover, the Momo framework is based on the SPS step-size, and we found it to be a better alternative to MomSPS across all considered tasks.
>
> [1] Orvieto \& Lin, An adaptive stochastic gradient method with non-negative gauss-newton stepsizes, arXiv preprint arXiv:2407.04358, 2024.

---

> > ### Author Response · Authors · 2024-11-21
> > **Rebuttals (Part 3)**
> >
> > **Q2:** This is a good point! Indeed, the theoretical advantage of NGN step-size is that it converges even if the interpolation condition does not hold while SPS${}\_{\max}$ algorithm converges up to interpolation error only. Moreover, we highlight that training a model with a huge number of parameters does not always mean that interpolation holds. In fact, when training large language models, the final loss is generally strictly greater than zero, even though the loss on individual batches is often very close to zero. This indicates that the interpolation condition does not hold for these types of problems. We refer to [1] for the scaling laws of Chinchilla models we use in our language modeling experiments. They empirically demonstrate that the loss is lower bounded by some constant.
> >
> > The main difficulty in the analysis of SPS${}\_{\max}$ and NGN comes from the fact that the step-size $\gamma_k$ and the stochastic gradient $\nabla f_{S_k}(x^k)$ are correlated. Therefore, upper bounding $\mathbb{E}[-\gamma_k \langle x^k-x^*, \nabla f_{S_k}(x^k) \rangle]$ becomes a challenging task. In the analysis of SPS${}\_{\max}$, they use a trivial lower bound on the SPS${}\_{\max}$ step-size which leads to looser inequality since the effective step-size can be significantly larger than this lower bound. This results in a worst-case analysis that ignores the adaptive effect of the step-size.
> > In contrast, we split the NGN step-size $\gamma_k$ into a deterministic and a stochastic part. In our analysis, this decomposition of the step-size $\gamma_k$ enables us to regulate the balance between the
> > descent term, which drives improvement in the objective, and the error term, which reflects possible
> > inaccuracies. More precisely, the descent term is weighted by a constant $c$ while the error term proportional to $\sigma^2_{\rm int}$ is weighted by $c^2$, which suggests that $c$ can be chosen to trade off the two terms to lead to the exact convergence similarly to the standard analysis of SGD.
> >
> > [1] Arora and Goyal, A theory for emergence of complex skills in language models, arXiv preprint arXiv:2307.15936, 2023.

---

> > > ### Comment · Reviewer_D8is · 2024-11-22
> > > **Reply to author response**
> > >
> > > Thanks for your response.
> > >
> > > - I appreciate your explanation in Part 1 regarding the comparison with Adam. I suggest incorporating it into the paper as well.
> > >
> > > - I reviewed the new experiment in Section G.9. Thank you for the effort. Unfortunately, it did not provide the expected intuition.
> > >
> > > - I also appreciate the discussion on the interpolation condition, and I am convinced that relaxing this condition is important.
> > >
> > > Overall, I thank you for your detailed response and would like to maintain my positive score for your paper.

---

> ### Author Response · Authors · 2024-11-23
> **Response to the reviewer**
>
> We thank the reviewer for their response. Regarding the second point, we would be grateful if the reviewer could provide further explanation regarding what he expected from the experiments in Section G.9. We clearly observe that NGN stepsize plays the role of the safeguard that is helpful and does not destroy performance. The method automatically detects if the effective step-size is too large and decreases it.

---

> > ### Comment · Reviewer_D8is · 2024-11-25
> > **Reply to the authors' response**
> >
> > I revisited this result and found it insightful. Figure 23 shows that with larger step-sizes, Adam tends to increase the learning rate in later iterations, whereas NGN-MDv1 maintains it more consistently. Initially, I had looked at Figure 22, where the general behavior of the methods is more similar.
> >
> > Thanks for your response.

---

### Official Review · Reviewer_sByP · 2024-11-04

**Soundness:** 3
**Presentation:** 3
**Contribution:** 2
**Rating:** 5
**Confidence:** 3

**Summary:**

This paper introduces a new optimization algorithm, NGN-M, which combines the NGN step size with momentum and develops NGN-MD, a coordinate-wise diagonal preconditioner version. The work aims to improve robustness in hyperparameter selection while maintaining state-of-the-art performance. The authors provide both theoretical and empirical evidence supporting the algorithm’s stability and convergence。

**Strengths:**

1. It is a natural and reasonable idea to extend the NGN method to the momentum setting. The authors achieved good empirical results and provided theoretical convergence guarantees.
2. The authors also introduced a diagonal step-size version, enriching the paper's content. The comparative experiments are comprehensive which improves over previous work.

**Weaknesses:**

1. Since the authors claim that the NGN-M algorithm can achieve better stability with larger step sizes, I would expect that the effective step size $\gamma_k$ during training dynamics would maintain a larger value, not just that the step size hyperparameter can be set larger. If it is merely the latter, wouldn't this imply a false sense of stability? In Figures 4 and 5, I do not see NGN-M demonstrating a consistently larger effective learning rate during training. If the observed stability is merely due to using a smaller effective step size relative to the hyperparameter, what is the practical significance of this stability?

2. I find the argument presented in Section 5.5 lacking rigor. The authors state, "Increasing the stepsize hyper-parameter of NGN-M leads to...," but this explanation is imprecise and somewhat misleading. The phenomenon of increasing the step size hyperparameter leading to convergence to flatter minima (i.e., with smaller eigenvalues of the loss Hessian) has already been explained by the EOS literature (see https://arxiv.org/abs/2103.00065). This body of work indicates that increasing the step size within a reasonable range generally leads to lower sharpness (i.e., reduced top eigenvalue) during neural network training. It is unclear if the authors' claim aligns with this well-known phenomenon, and they should reconsider this explanation.

Minor issues: I noticed that some notations are not precise.
1.	Ver.1 in Section 3.1 appears to have a typo. The term $\gamma^k$ in the lines 179 of the formula should be removed.
2.	I suggest the authors align the notation for iteration indices. Are the $\gamma$ values in lines 179 and 180 referring to the same iteration?

**Questions:**

1.	In addition to the questions posed in the Weaknesses1, I find that the comparison between NGN-MD and methods using a constant step size may not be entirely fair. Although the authors include a learning rate schedule in the comparative experiments after Section 5.3, it is evident that in Section 5.3, as shown in Figures 3 and 4, the advantage of NGN-MD becomes less pronounced compared to earlier experiments. I would be interested in seeing the stability comparisons in Sections 5.1 and 5.2 against classical optimizers that use warmup or learning rate schedules.

2.	Figure 16 is hard to understand. Does the loss along the top 2 eigenvectors truly represent the landscape along the update directions (such as negative gradient or various momentum directions)? And would a comparison under the same effective learning rate

---

> ### Author Response · Authors · 2024-11-21
> **Rebuttals**
>
> **W1:** We believe the reviewer interpreted “stability” differently from how we intended. Specifically,
> we do not mean that the algorithm employs large step-sizes. Effectively, our method gives an
> additional adaptivity boost compared to baselines. The method automatically detects if the training
> goes wrong and implements a lower effective step-size. It is an additional safeguard that is helpful
> and does not destroy performance.
>
> We emphasize that our focus is on stability from the
> perspective of step-size hyper-parameter tuning: the performance of our proposed algorithms is less
> sensitive to the choice of this hyper-parameter compared to baselines such as Adam. This notion
> of stability is the same as used in prior work (see references cited in the paper). Importantly, the
> optimal step-size hyper-parameter for Adam varies a lot from one task to another, especially when
> the domain of data changes (e.g., from vision tasks to NLP). This implies that determining the
> optimal step size in practice can be cumbersome for users, often requiring significant time and extra
> computations (which results in significant costs financially). From our experiments, NGN-MDv1
> requires less tuning of the step-size hyper-parameter. We will clarify what we exactly mean in the revised
> version. Moreover, we emphasize that the role of the step-size hyper-parameter $c$ in NGN-MDv1 is
> the maximum allowed effective step-size. We observe that the peak value of the effective step-size does
> increase when increasing $c$, and then it gradually decreases at the end of the training. These
> observations are in line with the learning rate schedulers that are used in practice.
>
> **W2:** Thank you for pointing this out. We will revise this section to make our explanation clearer. We address your comments in two parts:
>
> Regarding the connection to prior work: first, we agree that the observed phenomenon is strongly related to training at the Edge of Stability (EoS), as explored in [1] and other studies. However, we emphasize that [1] focuses on non-adaptive methods, both with and without momentum. The only work we are aware of that examines EoS behavior in adaptive methods is [2]. According to [2], Adam operates at an **adaptive** EoS (determined by the eigenvalues of the preconditioned Hessian), even as standard sharpness continues to increase throughout training. Our findings indicate that NGN-M operates at the Edge of Stability (EoS), despite employing adaptive step sizes. This discussion is reported in Section G.8 of the revised version of the paper.
>
> Regarding the current formulation in Section 5.5: We aimed to demonstrate why a large step-size hyperparameter does not negatively impact the performance of NGN-M. Specifically, SGDM fails to converge with a large step-size hyperparameter, diverging when operating beyond the EoS range. In contrast, NGN-M converges under the same step-size hyperparameter due to the adaptive nature of the NGN step size. We believe that the adaptivity of NGN-M enables it to operate effectively within the EoS range, even when the step-size hyperparameter is large. This is what we believe what makes NGN an interesting optimizer to consider in practice.
>
> [1] Cohen et al., Gradient descent on neural networks typically occurs at the edge of stability, arXiv preprint arXiv:2103.00065, 2021.
>
> [2] Cohen et al., Adaptive gradient methods at the edge of stability, arXiv preprint arXiv:2207.14484, 2022.
>
> **W3:**  We thank the reviewer for pointing out the typo. We also changed the iteration counter of the momentum term $m^k$, so that the new iterate $x^{k+1}$ is computed based on the terms from iteration $k$ or before: $x^k, m^{k-1}, \gamma_k, \nabla f_{S_k}(x^k)$. All these changes will be reported in the revised version of the paper.
>
> **Q1:** We would like to clarify that we only compare NGN-MD against methods with adaptive diagonal step-size and momentum: Momo-Adam and Adam, in exactly the same hyper-parameter scheduling conditions. The only difference is in the particular way how each algorithm computes the adaptive step-size. Therefore, we believe that the provided set of experiments is fair as we compare how the particular choice of adaptive step-size influences the performance. The learning rate scheduler can be incorporated into any of the algorithms we consider. However, doing so may obscure the true differences in adaptive step-size behavior.
> Introducing a scheduler also adds another layer of complexity, as it requires separate tuning for each task. While there are guidelines available for selecting an appropriate scheduler, these choices may still be suboptimal so we feel this would make the result less trustworthy. We are happy to discuss further if the reviewer disagrees.

---

> > ### Author Response · Authors · 2024-11-21
> > **Rebuttals (Part 2)**
> >
> > **Q2:** Figure 16 is provided for illustration purposes. The loss surface along the top 2 eigenvectors represents the sharpness of the loss landscape at the final iterate. According to Figure 16, both SGDM and NGN-M tend to converge to flatter minima when increasing the step-size hyperparameter. The main difference between the two algorithms is in the fact that NGN-M adapts to the curvature around the current point and therefore can work even with a large step-size hyperparameter.

---

> ### Author Response · Authors · 2024-11-23
> **Reminder**
>
> Dear reviewer,
>
> We would like to remind you that the discussion period ends soon. Therefore, we would like to know if there are concerns left unaddressed or needed to be clarified. We would be happy to discuss them further.

---

### Author Response · Authors · 2024-11-21
**General response to all reviewers**

We thank the reviewers very much for their dedication to the reviewing process and for taking the time to carefully study our work. We provide detailed responses to all raised concerns and questions. Moreover, we made several changes to the paper that we highlighted in blue color. We refer to the additional/revised sections in the response to each reviewer separately.

---

### Author Response · Authors · 2024-12-04
**Discussion period summary**

Dear reviewers,

We hope our rebuttals answered all your concerns. Now we would like to summarize the discussion period of the review process.

1. The reviewers acknowledged the importance of the theoretical analysis of the proposed methods. They found the analysis to be rigorous and clear. In particular, we provided $\mathcal{O}(1/\sqrt{K})$ convergence of NGN-M algorithm under standard assumptions improving previous results of adaptive algorithms with momentum without the requirement of bounded gradients. We explained the main difficulty in analyzing adaptive algorithms and showed how we managed to overcome this issue for NGN-M algorithm.
2. The reviewers highlighted the extensive empirical validation of the proposed methods on image and language modeling tasks. In particular, they emphasized that the proposed algorithms improve the stepsize resilience property over baselines such as SGDM, Momo, and Adam.
3. We explained that computing a step of NGN-MD algorithms does not significantly increase computation time and can be efficiently implemented in practice.
4. We provided additional experiments on NLP tasks (training LSTM and Transformer models) comparing NGN-M against Momo, SGDM, and NGN. We demonstrate that NGN-M achieves the improved stepsize resilience property in these tasks as well.
5. We compared the adaptive stepsize of NGN-MDv1, Momo-Adam, and Adam optimizers in training Resnet20 and ViT models. The results suggest that the adaptive step-size of NGN-MDv1 is more conservative and does not allow the effective step-size to increase too much even when the step-size hyper-parameter is set to be large. This demonstrates that the NGN-MDv1 is less sensitive to the choice of the step-size hyper-parameter while allowing it to reach comparable or superior performance to other optimizers.

Best regards,

Authors

---

### Meta-Review · Area_Chair_H4xJ · 2024-12-20

**Metareview:**

This paper considers a new step size called NGN proposed in a previous work and integrates into momentum method. The previous work only considers it for SGD update. The main contribution of this paper is the analysis of the momentum method (ie heavy-ball) method using the NGN step size.

The paper receives 5, 5, 8, 6 scores giving an average score of 6 leaning towards the acceptance. However, during the discussion period, a concern was raised about the results. In particular, the result of proving the momentum method requires the momentum parameter to decrease to zero ($\beta = \lambda/(1+\lambda)$, and $\lambda\leq c \rightarrow 0$ . This is a critical issue of this result as it is not the standard momentum method where $\beta$ is close to 1 or usually is a constant. Making $\beta$ go to zero make the method close to SGD, which is less interesting. Given this concern, AC decides to reject the paper and encourage the authors to solve this issue in order to claim a contribution for momentum methods with NGN step size.

**Additional Comments On Reviewer Discussion:**

Several reviewers engaged with the authors during the author reviewer discussion period and acknowledge the efforts of the authors to address their concerns during their initial reviews. However,  during the discussion period, there is a new concern raised about the setting of the momentum parameter. This is a critical concern of this paper that leads to the decision of rejection.

---

### Decision · Program_Chairs · 2025-01-22

Reject